# VIRTUE: Visual-Interactive Text-Image Universal Embedder

**Wei-Yao Wang**[♠][*]**, Kazuya Tateishi**[♠]**, Qiyu Wu**[♠]**, Shusuke Takahashi**[♠]**, Yuki Mitsufuji**[♠,♣]
[♠]Sony Group Corporation, [♣]Sony AI
{first_name.last_name}@sony.com

https://sony.github.io/virtue/

## Abstract

Multimodal representation learning models have demonstrated successful operation across complex tasks, and the integration of vision-language models (VLMs) has further enabled embedding models with instruction-following capabilities. However, existing embedding models lack visual-interactive capabilities to specify regions of interest from users (e.g., point, bounding box, mask), which have been explored in generative models to broaden their human-interactive applicability. Equipping embedding models with visual interactions not only would unlock new applications with localized grounding of user intent, which remains unexplored, but also enable the models to learn entity-level information within images to complement their global representations for conventional embedding tasks. In this paper, we propose a novel Visual-InteRactive Text-Image Universal Embedder (VIRTUE) that extends the capabilities of the segmentation model and the vision-language model to the realm of representation learning. In VIRTUE, the segmentation model can process visual prompts that pinpoint specific regions within an image, thereby enabling the embedder to handle complex and ambiguous scenarios more precisely. To evaluate the visual-interaction ability of VIRTUE, we introduce a large-scale Segmentation-and-Scene Caption Retrieval (SCaR) benchmark comprising 1M samples that aims to retrieve the text caption by jointly considering the entity with a specific object and image scene. VIRTUE consistently achieves a state-of-the-art performance with significant improvements across 36 universal MMEB (3.1%–8.5%) and five visual-interactive SCaR (15.2%–20.3%) tasks. The code, models, and benchmarks are available at https://github.com/sony/virtue.

## 1 Introduction

Embedding models have recently transitioned from two-tower architectures (e.g., CLIP (Radford et al., 2021), BLIP (Li et al., 2022a), SigLIP (Zhai et al., 2023)), which have been used for embedding-based evaluation (Girdhar et al., 2023) and cross-modal similarity matching (Hao et al., 2023; Han et al., 2024), to vision-language model (VLM)-based frameworks (e.g., GME (Zhang et al., 2025), LamRA (Liu et al., 2025b)) owing to VLMs' ability to ingest arbitrary combinations of textual and visual inputs into a single embedding space. Thanks to their inherent instruction-following capabilities, adopting VLMs as embedding models generalizes effectively across a wide range of zero-shot multimodal reasoning applications, including interactive information retrieval (Jiang et al., 2025b) and retrieval-augmented generation (Liu et al., 2025a).

Although VLM-based embedding models support interactive use, they rely on text as the primary human-machine interaction modality. In contrast, visual prompts, which have recently attracted attention in generative applications (You et al., 2024; Yuan et al., 2024; Lian et al., 2025), serve as an important but overlooked interaction channel. Visual prompting can not only enhance the downstream generation performance (Li et al., 2022b) but also provide precise spatial localization for fine-grained understanding (Liu et al., 2024c). This is particularly advantageous for embedding-based tasks, as it allows models to respond to visual inputs from the user beyond traditional global matching by capturing entity-level cues, thereby improving retrieval precision and alignment while complementing global representations.

---

[*]Project Lead.

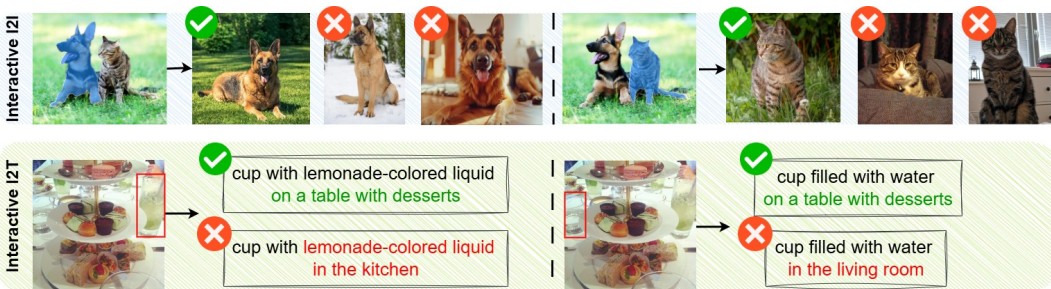

Figure 1: Visual-interactive paradigms for image-to-image (I2I) with masks assuming candidate images contain only dogs or only cats across different scenes, and image-to-text (I2T) with bounding boxes. False retrievals occur when retrieved content does not match the query's scene context.

Considering visual-interactive image-to-image (I2I) and image-to-text (I2T) retrieval scenarios, as shown in Fig. 1, where a user aims to retrieve different entities within the same image but under a shared global context, current embedding models rely solely on holistic image representations and fail to leverage explicit visual interactive signals (e.g., bounding boxes, points, and masks provided by users). As a result, they cannot isolate and retrieve the targeted entity while maintaining awareness of the broader scene (e.g., "grass" for the I2I scenario and "on a table with desserts" for the I2T retrieval). One possible strategy is to convert visual prompts into textual descriptions to guide retrieval; however, embedding models are not trained with spatially grounded supervision, which limits their ability to generalize to such interactive tasks. Another intuitive approach is to crop the region of interest (Subramanian et al., 2022), which can improve fine-grained understanding but sacrifices global contextual cues for compositional reasoning (e.g., understanding an object within the full scene as presented in Appendix E.4.1.). This limitation gives rise to a central challenge: *How can visual interaction capabilities be incorporated into embedding models, and how can we systematically evaluate their compositional reasoning on targeted image regions?*

In this paper, we propose VIRTUE, a visual-interactive text-image universal embedder that combines an off-the-shelf segmentation model (SAM2 (Ravi et al., 2025)) with a pretrained VLM to jointly encode entity- and global-level information from images and the textual descriptions. For visual-interactive scenarios, VIRTUE processes user-provided visual prompts by the prompt encoder within the segmentation model; for non-interactive scenarios, the prompt encoder is fed uniformly sampled points to produce a feature map composed of multiple entity-level information. The VLM then ingests arbitrary combinations of image and text embeddings, where each image embedding comprises both an entity-level embedding (from the segmentation model) and a global image embedding (from the VLM's vision encoder), and produces a single unified embedding for contrastive learning. In this manner, VIRTUE enables training on visual-interactive and non-visual-interactive data and supports entity-aware retrieval while preserving global scene context.

Since no existing benchmark evaluates visual-interactive embedding capabilities, we introduce 🦁SCaR, a large-scale Segmentation-and-Scene Caption Retrieval benchmark for visual-interactive image-to-text retrieval. In SCaR, an image together with a region of interest serves as a query, and the task is to retrieve the caption that describes the specified object in its global scene context. We constructed SCaR from five publicly available datasets: RefCOCO+ (Yu et al., 2016), RefCOCOg Mao et al. (2016), VisualGenome (Krishna et al., 2017), COCO-Stuff (Caesar et al., 2018), and ADE20k (Zhou et al., 2017). The annotations include images, bounding boxes, and captions that describe entities, relations, and the global scene context. To increase difficulties in reasoning, negative distractors are generated by replacing one of three elements of the ground-truth caption via prompting GPT-4V (OpenAI, 2023) instead of random sampling; for datasets that lack human captions (e.g., ADE20k), we generated ground-truth captions via carefully designed prompts to GPT-4V. To this end, SCaR comprises a vast collection of 1M samples that are divided into training and validation sets. A distinguishing characteristic of the proposed SCaR dataset is its ability to evaluate not only visual-interactive reasoning but also compositional scenarios, requiring models to perform fine-grained, context-aware cross-modal reasoning that goes beyond global image matching.

In summary, our contributions are three-fold:

- **Method Novelty:** We propose VIRTUE, a visual-interactive text-image universal embedder consisting of a VLM and a segmentation model to enable the visual interaction modality for human

interactions. The segmentation model allows users to optionally provide different types of visual prompts via its prompt encoder and reinforces VIRTUE to capture entity-level representations in addition to global context. We also conduct a systematic analysis of visual prompt integration to provide guidelines for enabling visual-interactive capabilities in embedding models.

- **Benchmark Novelty:** As there is no public visual-interactive embedding benchmark, we introduce SCaR, composed of 1M samples for visual-interactive image-to-text retrieval, to evaluate VIRTUE's capabilities. SCaR enables evaluation of advanced reasoning and compositional tasks in multimodal, visual-interaction-aware embedding scenarios that remain unexplored.
- **Experiment Novelty:** VIRTUE outperforms state-of-the-art embedding models on 36 MMEB tasks with significant gains from 3.1% to 8.5% and achieves improvements of 15.2% to 20.3% on five SCaR tasks, showing that equipping embedding models with visual-interactive capabilities benefits both visual-interactive and non-visual-interactive scenarios.

## 2 RELATED WORKS

**Multimodal Representation Learning.** Early progress in text-image representation learning was driven by two-tower contrastive models that learn a joint embedding space by aligning an image encoder with a text encoder (e.g., CLIP (Radford et al., 2021), BLIP (Li et al., 2022a), SigLIP (Zhai et al., 2023), OpenCLIP (Cherti et al., 2023)). These models provide effective global image-text matching and have served as foundation models for building vision-language models (VLMs) that tackle zero-shot downstream tasks (Tong et al., 2024; Wang et al., 2025b; Deitke et al., 2025). Subsequent work has advanced embedding performance along various dimensions. For instance, UniIR (Wei et al., 2024) finetunes CLIP/BLIP with the late fusion of text and image embeddings, and UniME (Gu et al., 2025) distills text knowledge from an LLM followed by two-stage negatives contrastive learning. Magiclens (Zhang et al., 2024) incorporates open-ended instructions into dual-encoder architectures with training on large-scale instruction datasets. More recently, VLMs that accept arbitrary mixtures of visual and textual inputs and are trained with instruction-style objectives have emerged as flexible and unified embedding providers that better perform on multimodal reasoning and compositional queries, e.g., E5-V (Jiang et al., 2024), VLM2Vec (Jiang et al., 2025b), GME (Zhang et al., 2025), and LamRA (Liu et al., 2025b). Despite these advancements, existing embedding models only support textual instructions and lack native support for direct visual prompts; they are typically trained only on holistic image-text alignment (Oh et al., 2024). In contrast, VIRTUE integrates a segmentation model with a pretrained VLM to fuse segmentation-derived, prompt-conditioned entity representations with global representations, producing unified embeddings that are both entity-aware and context-preserving, as evaluated on both MMEB and our newly constructed SCaR.

**Interactive Embedding Benchmarks.** Since embedding models have shifted from uni-modal matching benchmarks (e.g., BEIR (Thakur et al., 2021), MTEB (Muennighoff et al., 2023)) to instruction-based cross-modal global matching, recent studies have introduced instruction-based multimodal benchmarks to probe the reasoning abilities of embedding models. M-BEIR (Wei et al., 2024) is a multimodal retrieval benchmark encompassing eight tasks from diverse domains using text instructions, while MMEB (Jiang et al., 2025b) extends the evaluation to 36 multimodal datasets covering classification, VQA, retrieval, and grounding tasks to assess instruction-following across multifaceted perspectives. While we utilize MMEB to evaluate our proposed method for universal embedding abilities, these benchmarks focus on text-based instructions and do not evaluate visual-interactive scenarios in which visual prompts are provided as inputs. To fill this gap, we introduce SCaR, a large-scale interactive image-to-text retrieval benchmark where each query consists of an image as well as a target bounding box, and the task is to retrieve captions that describe the specified entity within its global scene context. SCaR is composed of five visual-grounding and referring-expression datasets that test caption retrieval for a region-in-context, with negative distractors generated by GPT-4V to stress-test entity-in-context discrimination beyond the simple random negatives used in MMEB.

## 3 SCAR: SEGMENTATION-AND-SCENE CAPTION RETRIEVAL BENCHMARK

### 3.1 SCAR OVERVIEW

Current publicly available benchmarks primarily evaluate text instruction-following capabilities for embedding models. Although MMEB contains out-of-domain visual grounding tasks for RefCOCO (Kazemzadeh et al., 2014), it simplifies them by cropping the specified region as the target, thereby

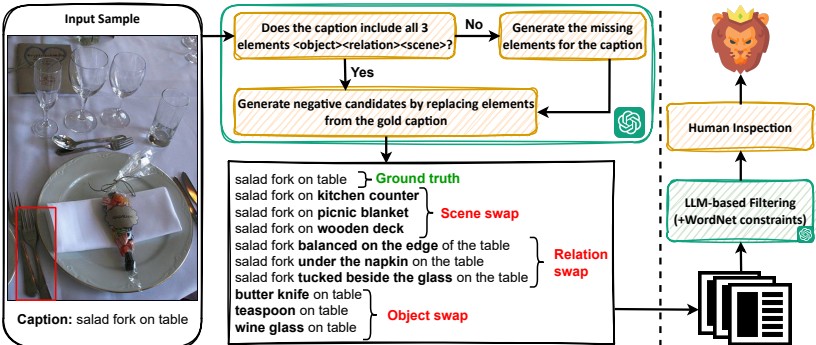

Figure 2: The data collection pipeline to build 🦁SCaR. We adopt GPT-4V to generate missing elements for the ground-truth caption as well as negative candidates. Collected samples (left) are filtered via LLM-then-human inspection (right) to ensure quality. Each SCaR sample contains an image with a bounding box, one ground-truth caption, and nine distractors.

neglecting the broader scene context within an image. Consequently, existing embedding models struggle with inputs that include visual regions of interest in visual-interactive retrieval tasks. To address this limitation, we introduce 🦁SCaR, a segmentation-and-scene caption retrieval benchmark that challenges models with reasoning and compositionality for text caption retrieval based on a given image and a specified region. SCaR comprises images, segmentations with bounding boxes, and text captions from five public datasets: RefCOCO+ (Yu et al., 2016), RefCOCOg (Mao et al., 2016), VisualGenome (Krishna et al., 2017), COCO-Stuff (Caesar et al., 2018), and ADE20k (Zhou et al., 2017). Distinct from existing benchmarks, SCaR provides multiple negative candidates per sample, which are generated by GPT-4V (OpenAI, 2023) through element-swapping in the ground-truth caption with false replacements, forming a large-scale benchmark of 1M samples covering diverse applicability.

**Task Definition.** The main difference between conventional image-to-text retrieval (e.g., MSCOCO_i2t in MMEB) and visual-interactive image-to-text retrieval is the additional region-of-interest input[1]. Formally, given an input image $I$ and a bounding box $P = [x_{min}, y_{min}, x_{width}, y_{height}]$ with ten candidates $C = [c_1, \cdots, c_{10}]$, the goal is to find the most relevant text caption $t_{gt} = \mathrm{argmax}(\mathrm{sim}(\phi(I, P), \phi(C)))$, where $\phi$ is the embedding model and sim denotes cosine similarity. Since SCaR requires models to reason about both the specified object and the broader scene, candidate captions are intentionally challenging and demand joint reasoning over fine-grained bounding-box details and global image context.

### 3.2 COLLECTION PIPELINE

Fig. 2 illustrates the collection pipeline to construct SCaR, where all datasets are first converted to the COCO format to enable unified processing. To balance evaluation efficiency and coverage, we randomly sample up to five objects per image. For each sample, the image size, original caption, object category, and bounding box[2] are provided in the prompt template, as shown in Fig. 5. The prompt instructs GPT-4V to return a JSON object containing the ground-truth caption and nine negatives. Since some datasets lack complete descriptions in terms of `<object> <relation> <scene>` (e.g., ADE20k only provides object names, and some RefCOCO+ samples do not contain global scene context), the prompt asks GPT-4V to verify whether the caption contains all three elements, and to supplement any missing elements with careful and image-grounded descriptions.

Negative candidates are then generated via element-swapping from the gold caption. We define three swap strategies, each producing three candidates. 1) **Global scene swap**: Replace the scene

---

[1]Both retrieval scenarios include the text instruction (the MMEB instruction is shown in Tab. 10 of Jiang et al. (2025b)). SCaR uses the following instruction: "*Find the caption that best describes the segmented object, considering both local details and global context in the given image.*". For simplicity, we omit these instructions here.

[2]While some datasets provide segmentation masks, we observed that GPT-4V struggles to interpret them reliably. In contrast, bounding boxes are easier for GPT-4V to parse and align with prior prompt-based collection strategies (Wang et al., 2025c; Jiang et al., 2025a); therefore, we opt for bounding boxes for SCaR.

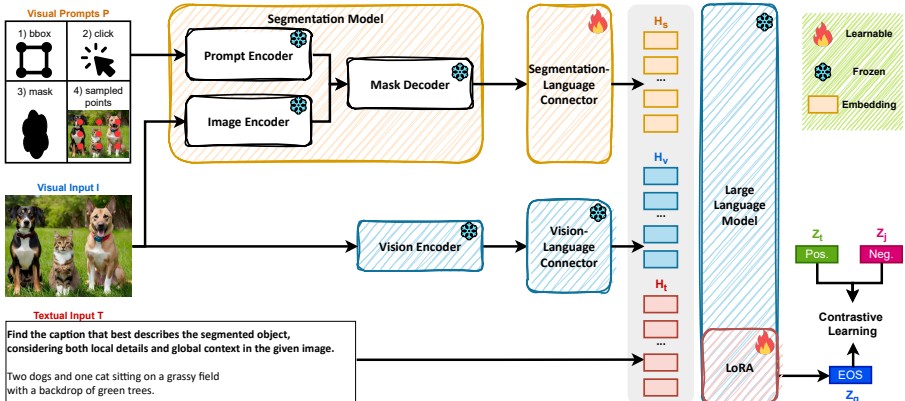

Figure 3: Overview of VIRTUE. The framework trained with contrastive loss consists of a segmentation model, a segmentation-language connector (orange), and a VLM (blue). It supports arbitrary combinations of visual and textual inputs with an optional visual prompt. If no prompt is provided, the model samples $N$ points uniformly from the image to extract entity-level information.

phrase with a clearly distinct environment (e.g., from "table" to "picnic blanket"). 2) **Relation swap**: Modify the relation of the specified object by borrowing from nearby objects or introducing hallucinated interactions (e.g., from "on the table" to "tucked beside the glass" or "under the napkin"). 3) **Object swap**: Substitute the target object with another object, either plausible but incorrect within the image or fabricated (e.g., from "salad fork" to "wine glass" or "butter knife"). In early trials, we observed that GPT-4V often generated ambiguous variants (e.g., "traffic light" vs. "stoplight", "zoo" vs. "safari park"), which could reduce the quality and reliability of the benchmark. To mitigate this, we explicitly add constraints to the instruction requiring swapped objects and scenes to belong to clearly distinct categories. We also encourage GPT-4V to produce diverse and creative negatives, ensuring coverage across different objects, relations, and scenes, while reducing the risk of overfitting to narrow patterns.

**LLM-then-Human Inspection.** Despite carefully designed prompts for guiding GPT-4V, some deviations from established rules result in the generation of subpar ground-truth and candidate generations, including generated ground-truth captions that lack scene context and negatives that involve synonymy or ambiguous objects and scenes. Therefore, we design a multi-stage filtering pipeline combining

Table 1: Dataset statistics of SCaR.

|  | Train | | Evaluation | |
| --- | --- | --- | --- | --- |
|  | #Images | #Annotations | #Images | #Annotations |
| RefCOCOg | 21,730 | 40,674 | 1,300 | 1,539 |
| RefCOCO+ | 16,847 | 38,807 | 1,500 | 2,764 |
| COCO-Stuff | 118,287 | 426,379 | 4,999 | 17,903 |
| VisualGenome | 86,414 | 357,583 | 5,000 | 15,571 |
| ADE20K | 20,210 | 94,271 | 2,000 | 9,368 |
| Total | 309,278 | 957,714 | 14,799 | 47,145 |

heuristic rules, LLM-based verification, and human inspection to improve dataset reliability and remove unethical samples. After the collection, GPT-4V is guided to verify whether each ground-truth caption contains all three elements `<object>` `<relation>` `<scene>` and to output these elements in a structured JSON format for both ground-truth and negative captions, as shown in Fig. 6. Then, WordNet (Miller, 1992) is applied to detect if any negative elements are synonyms of the corresponding gold elements. The sample is immediately discarded if any verification step fails. For the evaluation set, two independent human inspectors review all remaining samples with a focus on ambiguity, and remove any that are flagged by either inspector. For the training set, we assess quality indirectly by training models on it and evaluating them on the SCaR evaluation set (Sec. 5.3). This meticulous filtering regimen ensures the integrity and trustworthiness of the SCaR dataset, which comprises 957k training and 47k evaluation samples (see Tab. 1). More statistics and examples are provided in Appendix D.

## 4 VIRTUE

As illustrated in Fig. 3, since our goal is to create a general-purpose vision-language embedding model with the visual interaction capability, we propose **VIRTUE**, which consists of a VLM that processes visual and textual inputs as vision and language embeddings, respectively, and a segmentation model along with a segmentation-language connector that converts visual prompts (i.e.,

bounding boxes, masks, points, or sampled points when not explicitly provided) into the segmentation embeddings. Afterwards, the large language model (LLM) consumes the sequence of segmentation, vision, and language embeddings to generate the final input embedding using the final hidden state of the last token for contrastive learning.

## 4.1 CONVERTING VISUAL AND TEXTUAL INPUTS INTO EMBEDDINGS

Unlike the dual-tower structure of CLIP-based methods, VLMs incorporate a vision encoder, a vision-language connector, and an LLM, enabling inputs to be flexibly specified as unimodal (image or text) or bimodal (image-text pairs) within a shared representation space. The visual input streamline is processed by the vision encoder followed by the vision-language connector, both of which are components of the VLM, to produce $|v|$ global context vision embeddings $H_v \in \mathbb{R}^{|v| \cdot d} = \{h_v^1, \cdots, h_v^{|v|}\}$. Similarly, the textual streamline obtains $|t|$ textual embeddings $H_t \in \mathbb{R}^{|t| \cdot d} = \{h_t^1, \cdots, h_t^{|t|}\}$ through the text embedding layers of the LLM. To adapt the model to diverse downstream tasks, the query input further includes task definitions (e.g., "represent the given image with the following question") between the image and text, where the task-specific instructions are drawn from MMEB (Jiang et al., 2025b) as well as our proposed SCaR.

## 4.2 ENABLING VISUAL PROMPTS TO VLM EMBEDDERS

While existing VLM-based embedding models establish a general mechanism to accommodate diverse modality combinations, they often struggle to incorporate visual interaction prompts and may fail to capture fine-grained entity-level information. A common workaround is to crop bounding boxes, but cropping yields coarse rectangular regions and ineffective results (see Appendix E.1) that may include background, ignore inter-entity relations (e.g., "next to"), or cut across multiple entities. To address these limitations, we propose the visual prompt streamline, which supports bounding boxes, clicks (points), and masks as inputs. The visual prompt is then processed with the given image by the segmentation model to produce a segmentation map, which is subsequently passed through our proposed segmentation-language connector to generate segmentation embeddings.

**Segmentation Model.** We adopt the pretrained SAM-2 (Ravi et al., 2025) as the segmentation model within VIRTUE. A key advantage of using SAM-2 is that treating segmentation as an entity-level feature instead of relying on cropping provides a structured prior that aligns with human perception of discrete objects, producing features that more faithfully capture the semantics of the referenced entity. Specifically, SAM-2 includes a prompt encoder, an image encoder, and a mask decoder[3]. The image encoder processes the visual input to produce unconditioned feature embeddings, whereas the prompt encoder accepts visual prompts to define the extent of the object in an image. To cover conventional non-visual-interactive scenarios (e.g., MMEB), we set the visual prompt to $N$ uniformly sampled points when it is not explicitly provided to leverage SAM-2's inherent capability for automatic segmentation. This serves as a surrogate for user interactions and provides fine-grained entity-level cues in addition to the global context captured by the visual and textual streamlines. The mask decoder $f$ takes the outputs of the prompt encoder $E_p$ and the image encoder $E_i$ to produce a $64 \times 64$ segmentation feature map $F_s = f(E_p(P), E_i(I)) \in \mathbb{R}^{64 \cdot 64 \cdot d_s}$. The mask prediction, IoU, occlusion MLP heads, and upsampling process in SAM-2's mask decoder are discarded to utilize the segmentation feature map conditioned on visual prompts, since the segmentation feature map already encodes entity-level information from both the prompt and vision encoders; there is no need to merge multiple segmentation masks into a joint embedding space.

**Segmentation-Language Connector.** While we utilize the segmentation feature map $F_s$ to avoid the extra overhead of converting reconstructed masks into embeddings, directly flattening $F_s$ results in a sequence length of 4096, which cannot be feasibly processed when aligning to the LLM dimension $d$ due to GPU memory limitations. Therefore, we employ a 2D convolution layer $Conv2D$ to compress the feature map from 4096 tokens to $|S|$. The resulting representations are then projected via two MLP layers: first into $d_s$, and then into the LLM's hidden dimension $d$ for joint learning as:

$$H_s = MLP(Conv2D(F_s)) \in \mathbb{R}^{|S| \cdot d}. \tag{1}$$

---

[3]Visual prompts can be positive/negative for SAM-2, but we focus on positive prompts in this paper. We also omit the memory bank and memory attention designed for videos, as we focus on text and images.

## 4.3 TRAINING SCHEMA

The segmentation $H_s$, vision $H_v$, and text $H_t$ embeddings are concatenated in the order of segmentation-vision-text and subsequently fed into the LLM to generate the query embedding $\mathbf{z}_q$ or target embedding $\mathbf{z}_t$ from the final hidden state of the last token for contrastive learning. In this manner, the query embedding is encouraged to move closer to semantically similar targets while being pushed away from dissimilar ones, incorporating not only global matching signals but also entity-level information provided by the segmentation embeddings.

**Contrastive Learning.** Since VLMs are not originally tailored for representation learning, we adopt contrastive learning with the InfoNCE loss (van den Oord et al., 2018) on query embeddings $\mathbf{z}_q$ and target embeddings $\mathbf{z}_t$, each of which contains any combinations of $H_s$, $H_v$, and $H_t$. Formally, InfoNCE is applied over in-batch negatives:

$$\mathcal{L} = -\log \frac{\exp\left(\text{sim}(\mathbf{z}_q, \mathbf{z}_t)/\tau\right)}{\exp\left(\text{sim}(\mathbf{z}_q, \mathbf{z}_t)/\tau\right) + \sum\limits_{j=1}^{D} \exp\left(\text{sim}(\mathbf{z}_q, \mathbf{z}_j)/\tau\right)}, \qquad (2)$$

where $D$ denotes the set of hard negatives and $\tau$ is a temperature parameter. Following (Jiang et al., 2025b), GradCache (Gao et al., 2021) is employed to enable larger batch sizes, thereby improving the generalizability of the learned embeddings by leveraging a greater number of in-batch negatives.

## 5 EXPERIMENTS

### 5.1 EXPERIMENTAL SETUP

**Implementation Details.** We use Qwen2-VL-2B and Qwen2-VL-7B (Wang et al., 2024) as backbone VLMs for VIRTUE-2B and VIRTUE-7B, both with sam2.1_hiera_base_plus[4] as the segmentation model. To adapt the pretrained VLMs for embedding tasks, we apply LoRA (Hu et al., 2022) to the LLM within VIRTUE following (Jiang et al., 2025b), while training the segmentation-language connector from scratch. The segmentation embeddings are prepended to the inputs only when images are provided as inputs. The segmentation model, vision encoder, and vision-language connector are kept frozen to preserve the pretrained knowledge. VIRTUE is trained with 20 in-distribution MMEB-train datasets with a batch size of 1024 and LoRA rank of 8. Detailed settings are provided in Appendix C, with ablation and parameter studies in Appendices E.1 and E.2.

**Benchmarks.** We evaluate VIRTUE across 20 in-distribution test sets and 16 out-of-distribution test sets from MMEB (Jiang et al., 2025b) to assess its universal instruction-following embedding capabilities across diverse classification, VQA, retrieval, and visual grounding scenarios. To examine the visual-interactive capability, VIRTUE is evaluated on our proposed SCaR benchmark comprising five datasets with both out-of-domain and in-domain performance. Consistent with the MMEB benchmark, we report precision@1 across all experiments.

**Baselines.** We strive to provide a large number of comparisons against recent CLIP-based and VLM-based families. As CLIP-based methods, we compare against CLIP (Radford et al., 2021), BLIP2 (Li et al., 2023), SigLIP (Zhai et al., 2023), OpenCLIP (Cherti et al., 2023), UniIR (Wei et al., 2024), and Magiclens (Zhang et al., 2024). As VLM-based methods, we compare against E5-V (Jiang et al., 2024), GME (Zhang et al., 2025), MMRet (Zhou et al., 2025), LamRA (Liu et al., 2025b), VLM2Vec (Jiang et al., 2025b), and UniME (Gu et al., 2025). For fair comparisons, results on MMEB are reported from their corresponding papers, and all models on SCaR, except for +SCaR-train, are used off-the-shelf, with frozen weights. GME, LamRA, and VLM2Vec adopt 2B and 7B of Qwen2-VL (Wang et al., 2024), E5-V uses LLaVA-NeXT-8B (Liu et al., 2024b), and MMRet and UniME use LLaVA-1.6-7B (Liu et al., 2024a).

### 5.2 COMPARISON ON MMEB FOR UNIVERSAL EMBEDDING TASKS

Tab. 2 summarizes the overall performance of all methods, both with and without finetuning on MMEB-train. Our VIRTUE family consistently outperforms all baselines across the four core meta-task scenarios, as well as in both in-distribution (IND) and out-of-distribution (OOD) settings. Quan-

---

[4]https://github.com/facebookresearch/sam2

Table 2: Results on MMEB. The scores are averaged per meta-task. The improvements are calculated between VIRTUE and the best-performing 2B and 7B baselines. We highlight the **best** and second-best number of each column. Detailed scores are listed in Appendix E.7.

| | Per Meta-Task Score | | | | Average Score | | |
|---|---|---|---|---|---|---|---|
| | Classification | VQA | Retrieval | Grounding | IND | OOD | Overall |
| Model↓   \|   #Datasets → | 10 | 10 | 12 | 4 | 20 | 16 | 36 |
| **w/o Finetuning on MMEB-Train** | | | | | | | |
| CLIP$_{L/14}$(Radford et al., 2021) | 42.8 | 9.1 | 53.0 | 51.8 | 37.1 | 38.7 | 37.8 |
| BLIP2$_{opt-2.7b}$ (Li et al., 2023) | 27.0 | 4.2 | 33.9 | 47.0 | 25.3 | 25.1 | 25.2 |
| SigLIP$_{so400m-14-384}$ (Zhai et al., 2023) | 40.3 | 8.4 | 31.6 | 59.5 | 32.3 | 38.0 | 34.8 |
| OpenCLIP$_{L/14}$ (Cherti et al., 2023) | 47.8 | 10.9 | 52.3 | 53.3 | 39.3 | 40.2 | 39.7 |
| UniIR$_{CLIP\_CF}$ (Wei et al., 2024) | 44.3 | 16.2 | 61.8 | 65.3 | 47.1 | 41.7 | 44.7 |
| Magiclens$_{CLIP-L}$ (Zhang et al., 2024) | 38.8 | 8.3 | 35.4 | 26.0 | 31.0 | 23.7 | 27.8 |
| GME-2B (Zhang et al., 2025) | 54.4 | 29.9 | 66.9 | 55.5 | 49.2 | 55.2 | 51.9 |
| E5-V-8B (Jiang et al., 2024) | 21.8 | 4.9 | 11.5 | 19.0 | 14.9 | 11.5 | 13.3 |
| GME-7B (Zhang et al., 2025) | 57.7 | 34.7 | 71.2 | 59.3 | 53.6 | 58.8 | 56.0 |
| LamRA-7B (Liu et al., 2025b) | 59.2 | 26.5 | 70.0 | 62.7 | 53.0 | 55.4 | 54.1 |
| **w/ Finetuning on MMEB-Train** | | | | | | | |
| CLIP$_{L/14}$ (Radford et al., 2021) | 55.2 | 19.7 | 53.2 | 62.2 | 47.6 | 42.8 | 45.4 |
| OpenCLIP$_{L/14}$ (Cherti et al., 2023) | 56.0 | 21.9 | 55.4 | 64.1 | 50.5 | 43.1 | 47.2 |
| VLM2Vec-2B (Jiang et al., 2025b) | 58.7 | 49.3 | 65.0 | 72.9 | 64.9 | 53.3 | 59.7 |
| MMRet-7B (Zhou et al., 2025) | 56.0 | 57.4 | 69.9 | 83.6 | 68.0 | 59.1 | 64.1 |
| VLM2Vec-7B (Jiang et al., 2025b) | 62.7 | 56.9 | 69.4 | 82.2 | 71.4 | 58.1 | 65.5 |
| UniME-7B (Gu et al., 2025) | 60.6 | 52.9 | 67.9 | 85.1 | 68.4 | 57.9 | 66.6 |
| **VIRTUE (Ours) w/ Finetuning on MMEB-Train** | | | | | | | |
| VIRTUE-2B | 64.1 | 55.7 | 68.4 | 78.7 | 69.7 | 58.8 | 64.8 |
| VIRTUE-7B | **65.6** | **60.4** | **71.8** | **87.3** | **74.4** | **61.4** | **68.6** |
| Improvements (2B) | +5.4 | +6.4 | +1.5 | +5.8 | +4.8 | +3.6 | +5.1 |
| Improvements (7B) | +2.9 | +3.0 | +0.6 | +2.2 | +3.0 | +2.6 | +2.0 |

titatively, VIRTUE-2B achieves an average improvement of **5.1 points** over CLIP-based and other 2B models (from 59.7 to 64.8), while VIRTUE-7B surpasses existing 7B models with a **2.0-point** gain (from 66.6 to 68.6). While VLM-based models are superior to CLIP-based ones, VIRTUE significantly outperforms GME and LamRA with the same Qwen2-VL backbone, as well as MM-Ret and E5-V with different backbones across all meta-tasks, demonstrating the effectiveness and universality of VIRTUE. In addition, the comparisons between VIRTUE and VLM2Vec highlight the importance of incorporating entity-level information, since both models adopt the same training schema and data. The use of uniformly sampled points for non-visual-interactive tasks signifies that segmentation embeddings contribute positively to universal embedding performance by enriching global context with fine-grained object-level details. Compared to UniME-7B, VIRTUE-7B surpasses all meta-tasks scenarios, underscoring the generalized embedder ability of VIRTUE, which jointly integrates global context and segmentation-derived context with in-batch negatives.

## 5.3 COMPARISON ON SCAR FOR VISUAL-INTERACTIVE TASKS

To examine the visual-interactive capabilities, VIRTUE is compared with CLIP as well as the crop-based (ReCLIP (Subramanian et al., 2022)) and visual-hinting-based (explicitly add red circles to images (Shtedritski et al., 2023), denoted as +Red Circle) variants. VLM-based embedding models are compared with and without finetuning on MMEB-train, due to their effectiveness and task-following abilities. Since none of the baselines naturally accept bounding boxes as visual inputs, they are textualized as *Referring object bbox: {bbox}*, including after queries, while VIRTUE takes not only the textualized input but also visual prompts. In addition, we further conduct experiments by naively cropping the specified object using the corresponding bounding box as the visual input for all models (+Cropping); in this case, no textualized bounding box is required. To push the limits of visual-interactive capabilities and for fair comparisons, we further finetune VIRTUE, VLM2Vec, MMRet, and UniME on the MMEB-train checkpoints for 1k steps using a batch size of 1024 and a learning rate of 2e-6, following the aforementioned formats, which are denoted as +SCaR-train.

Tab. 3 reports the results under both out-of-domain and in-domain (i.e., +SCaR-train) scenarios. Without finetuning on MMEB-train, CLIP outperforms VLM-based models. In contrast, simply adding visual hints, which was observed effectively in (Shtedritski et al., 2023), performs slightly worse than CLIP, likely due to the more challenging nature of SCaR, where captions emphasize rea-

Table 3: Results on our proposed SCaR benchmark. All models incorporate bounding boxes in the textual prompt. +Cropping: Use only the cropped region of the image based on the given bounding box as input. +SCaR-train: Further finetune 1k steps with the SCaR training set.

| Model ↓ \| Datasets → | RefCOCO+ | RefCOCOg | VisualGenome | COCO-Stuff | ADE20K | Overall |
|---|---|---|---|---|---|---|
| | | | w/o Finetuning on MMEB-Train | | | |
| CLIP | 18.1 | 22.4 | 23.0 | 15.5 | 18.9 | 19.6 |
| ReCLIP | 17.6 | 23.0 | 21.3 | 12.6 | 15.9 | 18.1 |
| + Red Circle | 17.6 | 21.6 | 22.3 | 15.1 | 18.4 | 19.0 |
| GME-2B | 5.4 | 8.3 | 7.5 | 5.8 | 6.2 | 6.6 |
| +Cropping | 6.1 | 8.3 | 5.8 | 6.2 | 7.5 | 6.8 |
| GME-7B | 6.1 | 10.1 | 7.4 | 7.0 | 5.6 | 7.2 |
| +Cropping | 6.0 | 9.8 | 4.4 | 7.0 | 5.3 | 6.5 |
| LamRA-7B | 8.0 | 7.9 | 5.3 | 3.1 | 8.9 | 6.6 |
| +Cropping | 8.2 | 7.9 | 3.1 | 8.9 | 5.3 | 6.7 |
| | | | w/ Finetuning on MMEB-Train | | | |
| VLM2Vec-2B | 24.5 | 29.5 | 22.3 | 19.4 | 24.6 | 24.1 |
| +Cropping | 22.3 | 25.8 | 17.1 | 19.5 | 22.5 | 21.4 |
| +SCaR-train | 59.8 | 55.5 | 34.1 | 40.8 | 43.2 | 46.7 |
| VIRTUE-2B (Ours) | 28.8 | 42.4 | 24.4 | 29.9 | 27.5 | 30.4 |
| +Cropping | 24.4 | 36.3 | 14.2 | 25.8 | 23.4 | 24.8 |
| +SCaR-train | **64.2** | **65.3** | **41.4** | **54.2** | **56.0** | **56.2** |
| Δ (2B) | +4.3 | +12.9 | +2.1 | +10.5 | +2.9 | +6.3 |
| Δ (2B, +SCaR-train) | +4.4 | +9.8 | +7.3 | +13.4 | +12.8 | +9.5 |
| VLM2Vec-7B | 23.2 | 29.1 | 14.7 | 25.0 | 22.4 | 22.9 |
| +Cropping | 17.6 | 25.7 | 15.2 | 13.8 | 21.3 | 18.7 |
| +SCaR-train | 40.1 | 39.5 | 36.3 | 31.4 | 25.0 | 34.5 |
| MMRet-7B | 27.1 | 21.8 | 15.2 | 26.0 | 22.6 | 22.5 |
| +Cropping | 26.0 | 20.2 | 14.9 | 15.0 | 19.1 | 19.0 |
| +SCaR-train | 45.8 | 43.6 | 39.7 | 27.5 | 31.2 | 37.6 |
| UniME-7B | 31.4 | 32.8 | 19.0 | 25.3 | 23.0 | 26.3 |
| +Cropping | 25.8 | 24.0 | 15.2 | 24.6 | 22.3 | 22.4 |
| +SCaR-train | 57.8 | 59.3 | 44.5 | 41.9 | 43.6 | 49.4 |
| VIRTUE-7B (Ours) | 33.0 | 35.3 | 19.6 | 27.1 | 23.8 | 27.8 |
| +Cropping | 27.1 | 31.5 | 17.4 | 21.8 | 16.4 | 22.8 |
| +SCaR-train | **63.2** | **66.1** | **48.2** | **52.2** | **54.9** | **56.9** |
| Δ (7B) | +1.6 | +2.5 | +0.6 | +1.8 | +0.8 | +1.5 |
| Δ (7B, +SCaR-train) | +5.3 | +6.8 | +3.7 | +10.3 | +11.3 | +7.5 |

soning and compositionality. VLM-based models without finetuning from MMEB-train (i.e., GME, LamRA) are more prone to underperform compared to models with finetuning from MMEB-train (i.e., VLM2Vec, MMRet, UniME), which can be attributed to the visual grounding datasets within MMEB-train that crop the ground-truth region as the target. Meanwhile, our proposed VIRTUE improves by **6.3** and **1.5 points** on average for the 2B and 7B models, respectively, which implies that conditioning embeddings on visual prompts facilitates the learning of fine-grained representations for regions of interest. Although directly cropping the specified object enables a fine-grained understanding, the detrimental effects of +Cropping across all models and ReCLIP verify that it sacrifices global scene information for compositional reasoning. The results of +SCaR-train show that further finetuning with our collected SCaR training set is able to boost visual-interactive reasoning even for models that do not originally support visual prompts. Nonetheless, VIRTUE, equipped with the segmentation streamline, leads to a larger performance gain of **9.5 points** for the 2B model and **7.5 points** for the 7B model, which demonstrates the effectiveness of SCaR-train and incorporating visual prompts for not only fine-grained information but also user-enabled visual interactions. We include additional reevaluations on MMEB and qualitative results on SCaR with SCaR-train in Appendices E.3 and E.5, respectively. More application paradigms are presented in Appendix E.4.

## 5.4 IMPACTS OF VARIOUS VISUAL PROMPTS ON VIRTUE

To investigate the impacts of various visual prompts on VIRTUE, we analyze the robustness of VIRTUE-7B and VIRTUE-7B-SCaR-train under noisy and misaligned bounding boxes from the SCaR evaluation set. We further perform SCaR evaluations by replacing ground-truth bounding boxes with centroid points or random boxes, examining whether VIRTUE can effectively handle different types of interactive visual prompts for embedding generation. Finally, on MMEB, we study the effect of removing uniformly sampled points, either during inference only (w/o inference) or during both training and inference (w/o all), to understand how reliance on uniform point prompts influences performance.

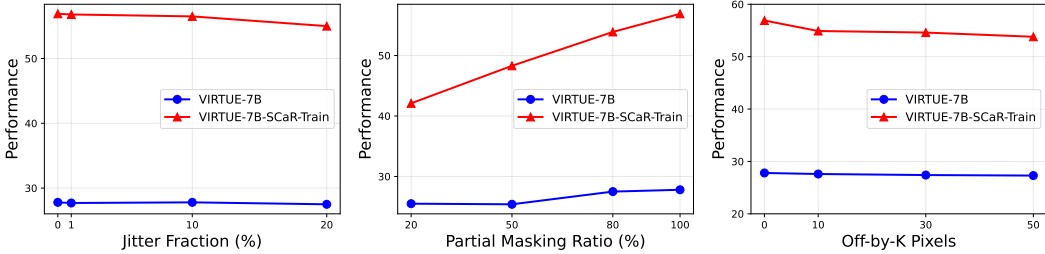

Figure 4: Analysis of noise and misalignment in visual prompts on SCaR: jittered boxes (left), partial masks (middle), and off-by-k pixels (right).

Table 4: Analysis on SCaR using points and randomly generated bounding boxes.

| | VIRTUE-7B | | | VIRTUE-7B-SCaR-train | | |
|---|---|---|---|---|---|---|
| | bbox | random | points | bbox | random | points |
| SCaR | 27.8 | 9.2 | 26.6 | 56.9 | 13.0 | 45.0 |

Table 5: Effect of removing uniformly sampled points during inference (w/o inference) or both training and inference (w/o all) on MMEB. Results using uniform points are shown in bold.

| | VIRTUE-7B | | | VIRTUE-7B-SCaR-train | | |
|---|---|---|---|---|---|---|
| | uniform | w/o inference | w/o all | uniform | w/o inference | w/o all |
| MMEB | **68.6** | 24.1 | 65.5 | **66.8** | 41.9 | 43.4 |

**Robustness to Noisy or Misaligned Visual Prompts.** Fig. 4 presents results with different fractions of jittered bounding boxes, varying mask ratios for partial boxes, and off-by-k pixels. Both VIRTUE-7B and VIRTUE-7B-SCaR-train remain stable across different amounts of jitter and off-by-k offsets, demonstrating robustness of the segmentation streamline. On the other hand, using only 20% partial masks results for VIRTUE-7B-SCaR-train remains superior to baselines like VLM2Vec-7B-SCaR-train, suggesting that our model remains robust even when visual prompts provide limited coverage.

**Effect of Using Points vs. (Random) Bounding Boxes.** To understand if VIRTUE takes advantage of segmentation tokens, we hypothesize that if the model effectively leverages segmentation tokens, randomly sampling visual prompts should degrade performance; if it does not, performance should remain unchanged or improve (Makansi et al., 2022; Wang et al., 2025a). Tab. 4 shows the SCaR performance when bounding boxes (bbox), randomly sampled boxes (random), or centroid points (points) are adopted as visual prompts. Using points achieves performance comparable to bounding boxes for VIRTUE-7B, while VIRTUE-7B-SCaR-train exhibits a larger drop, indicating that fine-tuning on SCaR encourages a stronger reliance on bounding box prompts. The severe degradation with random boxes for both models further confirms that VIRTUE meaningfully uses the provided visual prompts when generating embeddings.

**Impact of Removing Uniformly Sampled Points.** Since VIRTUE incorporates uniformly sampled points as visual prompts for non-visual-interactive scenarios, Tab. 5 reports results obtained by removing these points either only during inference (w/o inference) or during both training and inference (w/o all). The performance declines in both settings indicate that uniform points help the segmentation streamline capture entity-level structure, which complements global context and leads to more fine-grained embeddings.

# 6 CONCLUSION

In this paper, we proposed VIRTUE, a novel visual-interactive text-image universal embedder for both instruction-following and visual-interactive embedding scenarios. Distinct from existing models that support only text as the interaction modality, VIRTUE equipped with pre-trained SAM2 as the segmentation model is able to process visual prompts as inputs, enabling the model to jointly capture global context as well as entity-level information. We constructed SCaR, a large-scale segmentation-and-scene image-to-text retrieval benchmark, to assess VIRTUE's visual-interaction reasoning abilities, which consists of 1M samples with challenging negative candidates generated by replacing elements of the ground-truth captions. Experiments on both MMEB and SCaR demonstrate that VIRTUE is consistently superior to state-of-the-art approaches by 3.1%-8.5% and 15.2%-20.3%, respectively. We believe VIRTUE serves as a generic framework for conventional instruction-following and visual-interactive embedding tasks, and that SCaR opens up new human-AI interaction opportunities for embedding models. We present further discussion of limitations and broader impacts in Appendices A and B.

## ACKNOWLEDGEMENTS

We gratefully acknowledge the support of all those who contributed to this research. We thank Naofumi Akimoto and Toshimitsu Uesaka for their valuable feedback and insightful suggestions that helped refine this work.

## ETHICS STATEMENTS

The SCaR dataset is constructed from publicly available datasets (RefCOCO+, RefCOCOg, VisualGenome, COCO-Stuff, and ADE20K), which mitigates potential privacy concerns and harmful content. The dataset is primarily generated using GPT-4V with subsequent human inspection to remove ethically problematic content. While GPT-4V may inherit biases from its training data, our prompt design explicitly encourages diverse and creative captions, and a comprehensive filtering pipeline is applied to examine the generated captions. This design reduces human annotation bias and avoids repetitive lexical overlap, as illustrated in Fig. 7. Moreover, although VIRTUE builds upon a pre-trained VLM, it is further fine-tuned into an embedding model, which significantly minimizes the risk of harmful or copyrighted content generation while preserving the semantic richness needed for downstream tasks.

**LLM Usage.** In addition to GPT-4V used for building the SCaR benchmark, we use LLM solely for polishing the manuscript.

## REPRODUCIBILITY STATEMENTS

The codebase, VIRTUE models, and SCaR benchmark samples are provided at https://github.com/sony/virtue to advance the community with the realm of interactive representation learning, and the experimental details as well as model configurations are summarized in Sec. 5.1 and Appendix C. Evaluations on the MMEB benchmark follows their official use, while SCaR is detailed in Sec. 3 and Appendix D.

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

In the Appendix, Sec. A states the ethical consideration and limitations of this paper, Sec. B discusses broader impacts for this work, Sec. C summarizes the implementation configurations, and Sec. D presents additional details for our proposed SCaR, including the prompt templates (Secs. D.2 and D.3) and additional statistics (Sec. D.4). We also include extensive experiments in Sec. E, including an ablation study (Sec. E.1), hyperparameter study (Sec. E.2), reevaluation on MMEB with SCaR-train (Sec. E.3), application paradigms on enabling VIRTUE with visual interactions (Sec. E.4), and qualitative results on SCaR (Secs. E.5 and E.7).

## A    LIMITATIONS AND FUTURE DIRECTIONS

While our approach represents a significant step toward reinforcing visual-interactive capabilities in embedding models, the main limitation lies in training VIRTUE only with MMEB and SCaR, primarily due to computational constraints. Prior work (e.g., (Zhou et al., 2025)) has shown that incorporating more diverse datasets can further improve universality, suggesting that expanding our training sources would yield additional performance gains. Additionally, our evaluation emphasizes interactive image-to-text retrieval (via SCaR) as a primary validation task. While this provides strong evidence for the effectiveness of our approach, we only cover interactive image-to-image retrieval as case studies in Sec. E.4.1, largely due to copyright and ethical concerns associated with constructing suitable benchmarks. Nevertheless, this remains an important and complementary dimension for assessing visual-interactive embedding models. These limitations reflect our overarching objective: to pave the way for embedding models that are not only text-interactive but also visually interactive, while maintaining a strong universal embedding performance. In future work, we aim to extend our framework with more advanced training strategies and diverse as well as multifaceted datasets, including safe and ethically curated setups for interactive image-to-image retrieval, to fully unlock the potential of visual-interactive embedding models.

## B    BROADER IMPACTS

As VIRTUE introduces a new paradigm for visual-interactive embedding models, it has the potential to serve as a multimodal encoder that naturally supports visual prompts for VLMs, in contrast to post-hoc finetuning approaches (e.g., (You et al., 2024; Lian et al., 2025)). It may also benefit the development of multimodal foundation models built on top of CLIP (e.g., video-text-to-audio generation models (Cheng et al., 2025), text-to-image generation models (Rombach et al., 2022)), given VIRTUE's superior performance over existing CLIP-based methods. Moreover, embedding-based evaluation methods such as CLIPScore (Hessel et al., 2021) are widely used as automatic and reference-free metrics for holistic image-text alignment. VIRTUE could advance this line of evaluation by serving as a stronger embedding-based standard: while CLIPScore represents a subset functionality, VIRTUE further enables region-specific interactions in addition to capturing global context to compute text-image similarities.

## C    IMPLEMENTATION DETAILS

VIRTUE-2B and VIRTUE-7B are trained with the $\tau$ of 0.02 and learning rate of 2e-5, and input resolution of $1344 \times 1344$. The lengths of vision and text embeddings default to VLMs. Since $|S|$ segmentation embeddings are prepended to the sequence, their attention masks and positional encodings are set sequentially, following the original VLM design. $d_s$ is 256 from SAM2.1's default configuration, and $d$ is 1536 for 2B and 3584 for 7B from Qwen2-VL's default settings. For evaluating MMEB, VIRTUE is trained with MMEB-train with 5k steps, and is further trained with 1k steps with SCaR-train for (+SCaR-train in Tab. 3). The VIRTUE 2B and 7B training was conducted on 8×H100 80GB, where MMEB-train took around 74 and 189 hours, respectively, and SCaR-train took around 12 and 30 hours, respectively. The kernel size and stride of $Conv2D$, the number of sampled points $N$, and the number of segmentation embeddings $|S|$ are empirically set to 4, 9, and 256, respectively. Detailed configurations are summarized in Tab. 6.

Table 6: Detailed configurations for VIRTUE 2B and 7B models.

| Configurations | VIRTUE-2B | VIRTUE-7B |
|---|---|---|
| **MMEB-Train** | | |
| VLM | Qwen2-VL-2B | Qwen2-VL-7B |
| Segmentation model | SAM2.1-hiera-base-plus | |
| Image resolution | 1344×1344 | |
| Segmentation-language connector | Random initialized | |
| LoRA rank | 8 | |
| LoRA dropout | 0.1 | |
| LoRA alpha | 64 | |
| Temperature $\tau$ | 0.02 | |
| Batch size | 1024 | |
| Training steps | 5000 | |
| Warmup steps | 200 | |
| Learning rate | 2e-5 | |
| $d_s$ | 256 | |
| $d$ | 1536 | 3584 |
| Sampled points $N$ | 9 | |
| # Segmentation embeddings $|S|$ | 256 | |
| GPU | 8×H100 80G | |
| Precision | bf16 | |
| Optimizer | AdamW ($\beta_1 = 0.9, \beta_2 = 0.999$) | |
| Training time | 74 hours | 189 hours |
| **SCaR-Train** | | |
| VLM | Checkpoint of VIRTUE-2B | Checkpoint of VIRUTE-7B |
| Segmentation model | SAM2.1-hiera-base-plus | |
| Image resolution | 1344×1344 | |
| Segmentation-language connector | Checkpoint of VIRTUE-2B | Checkpoint of VIRUTE-7B |
| LoRA rank | 8 | |
| LoRA dropout | 0.1 | |
| LoRA alpha | 64 | |
| Temperature $\tau$ | 0.02 | |
| Batch size | 1024 | |
| Training steps | 1000 | |
| Warmup steps | 100 | |
| Learning rate | 2e-6 | |
| $d_s$ | 256 | |
| $d$ | 1536 | 3584 |
| Sampled points $N$ | 9 | |
| # Segmentation embeddings $|S|$ | 256 | |
| GPU | 8×H100 80G | |
| Precision | bf16 | |
| Optimizer | AdamW ($\beta_1 = 0.9, \beta_2 = 0.999$) | |
| Training time | 12 hours | 30 hours |

# D   SCaR Details

## D.1   Dataset Details

We summarize the five datasets used for building SCaR below:

- **RefCOCO+** (Yu et al., 2016): A large-scale dataset for referring expression comprehension and segmentation, containing around 45k expressions over 19k images from MS-COCO. Crucially, it prohibits location-based words (e.g., "left", "right"), forcing models to rely on appearance and contextual reasoning–a more robust form of vision-language grounding that moves beyond simple spatial relationships. Its unique constraints make it a strong test of a model's ability to truly understand an object's visual properties relative to its surroundings.
- **RefCOCOg** (Mao et al., 2016): A referring expression dataset with around 50k expressions for around 30k images, collected with longer and more descriptive annotations. This rich

Table 7: Inter-annotator agreement and the number (and proportion) of dropped samples from the original SCaR evaluation set. We report the average Cohen's Kappa between the two annotators.

| Cohen's Kappa | Drop samples (both annotators flagged) | Drop samples (either annotator flagged) |
|---|---|---|
| 0.89 | 4021 (7.7%) | 1638 (3.1%) |

linguistic detail makes it a standard benchmark for evaluating fine-grained vision-language grounding, as it requires models to reason over nuanced and complex natural language descriptions rather than just a few keywords.

- **VisualGenome** (Krishna et al., 2017): A massive vision-language dataset that provides dense annotations for over 100k images, including objects, attributes, and region-level descriptions. Its most notable feature is the inclusion of scene graph annotations, which explicitly model relationships between objects. This structured data is invaluable for training and evaluating models on compositional reasoning, enabling a deeper understanding of complex scenes beyond simple object detection.

- **COCO-Stuff** (Caesar et al., 2018): An extension of the MS-COCO dataset with over 160k images annotated for 91 "stuff" classes (e.g., sky, grass) in addition to the existing "thing" categories. This comprehensive annotation scheme makes it a primary benchmark for semantic segmentation and tasks that require holistic scene understanding, as it allows models to learn the fine-grained contextual relationships between foreground objects and their background environments.

- **ADE20K** (Zhou et al., 2017): A challenging scene parsing dataset with 20k images and 150 fine-grained semantic categories. The dataset's diversity, spanning a wide range of indoor and outdoor scenes, coupled with its high-quality pixel-level labels, makes it an essential standard for evaluating the performance of semantic segmentation and general scene understanding models.

### D.2 PROMPT TEMPLATE FOR BUILDING SCAR

Fig. 5 illustrates the detailed prompt template used for building the SCaR dataset (Sec. 3.2), where each sample varies in image size, caption, category, and bounding box (bbox) as provided in the original datasets. The ground-truth instruction guides GPT-4V to determine whether the three elements `<object> <relation> <scene>` are satisfied; if any element is missing, GPT-4V is required to complete it. Subsequently, the negatives instruction directs GPT-4V to replace each element from the ground-truth caption. Notably, we explicitly include creativity and diversity instructions to encourage GPT-4V to generate more diverse and imaginative negatives, as visualized in the word clouds in Fig. 7. To ensure GPT-4V follows these instructions, we provide an example output at the end of the prompt.

### D.3 PROMPT TEMPLATE FOR LLM-BASED FILTERING

Fig. 6 shows the prompt for LLM-based filtering. Generally, we adopt GPT-4V to guide the data verification from multifaceted perspectives. If all conditions are passed, the output JSON object contains the status field with "passed" as well as the extracted objects, relations, and scenes for both ground-truth and negative captions, which are then passed on to WordNet to inspect again in terms of semantic differences.

### D.4 ADDITIONAL SCAR STATISTICS

We further analyze the composition of SCaR to better understand its benchmark characteristics. Tab. 7 presents the inter-annotator agreement and the number of dropped samples in the SCaR evaluation set. The high agreement score indicates that the two independent annotators were highly consistent in the dataset inspection process. After inspection, we removed approximately 11% of the samples: 7.7% were flagged by both annotators, and an additional 3.1% were flagged by either annotator.

Fig. 7 summarizes detailed statistics including word clouds, annotation counts, and image distributions for both the training and evaluation splits. Attributed to the creative and diverse instruction

**Prompt Template for Collecting SCaR with GPT-4V**

You are an AI visual assistant capable of correctly analyzing a single image. You receive the specific object locations within the image, along with detailed coordinates. These coordinates are in the form of bounding boxes, represented as (x1, y1, x2, y2). These values correspond to the top left x, top left y, bottom right x, bottom right y. The height and width of the image you receive are 427 and 640, respectively. The global caption, category names, and bounding box coordinates of objects are as follows:
```
'''
caption: An egg salad sandwich with an orange toothpick holding it together.
category: food
bbox: [135.57, 248.43, 157.89, 278.22]
'''
```
Your job is to output plain JSON only, with no Markdown or code fences, defining:
1. ground_truth
   - Full caption: If 'caption' already contains an object, a clear relation, and a scene/context phrase (e.g., "Dog sleeping on the sofa"), reuse it verbatim.
   - Relation-only: If 'caption' contains an object and relation but no scene (e.g., "Dog lying on the rug"), append a concise scene descriptor observed in the image (e.g., "Dog lying on the rug in the living room").
   - Object-only: If 'caption' is just a bare object label (e.g., "boat"), generate a concise `<object> <relation> <scene>` description from the image:
      1. Start with the exact object label.
      2. Describe its visible appearance or action (relation).
      3. Add a simple scene context based strictly on what you see (scene).
   - Never rewrite or paraphrase a full or relation-only caption beyond adding the missing scene; do not invent new objects or actions.
2. negatives: an array of exactly 9 caption objects, each with:
   - 'text': the caption string.
   - 'type': one of ['global_context', 'background_relation', 'object_swap'].
Use these rules for negatives:
- global_context
   - Identify the scene phrase in 'ground_truth'. Replace it with a different, clearly distinct scene, not a sibling or near-synonym (e.g., don't swap "zoo" with "safari park").
   - Choose contexts that are plausible but semantically distant (e.g., "in the kitchen" vs. "on the beach", not "in the playground" vs. "on the sports field").
- background_relation
   - Keep the same main object and the scene phrase from 'ground_truth'.
   - Change its relation in a creative way—feel free to introduce a novel interaction or action, even if it involves an object or element not explicitly listed among the nearby objects.
   - Ensure each relation is creative and diverse, and deliberately false when applied to the main object and its scene from the given image.
- object_swap
   - object_swap: swap out the object class for a different one, keeping the same relation to the scene or nearby object (e.g., "Chair with lamp between the two beds"). Do not replace with synonyms or hyponyms/hypernyms. Avoid changes like "girl" to "woman", or "traffic light" to "stoplight". These are lexical variants, not distinct categories. Ensure the swapped object is real-world valid and contextually plausible. Ensure swapped objects are from a clearly distinct category (not mere sibling classes or near-synonyms). Avoid replacing an object with another from the same fine-grained group (e.g., don't swap "bowl" with "mug" or "plate"; "armchair" with "sofa").
   - Vary the swapped classes–avoid reusing "cat", "dog", or "chair" in every example.
Creativity & Diversity Constraint
- Do not only ensure uniqueness; actively push for variety within each negative type so that your negatives are diverse and creative, forcing models to use richer context signals.
Make sure the 9 'negatives' contain exactly 3 hard negatives for each type ('global_context', 'background_relation', 'object_swap'). Ensure all negatives are challenging–they should look plausible when cropping to the mask, but only resolvable with full-image context and mask identity (i.e., a naive crop-only baseline would confuse it with the GT). Don't add comments in the JSON file.
Example (abbreviated) output:
```
{
  "ground_truth": "Traffic light above the crosswalk",
  "negatives": [
    "text": "Traffic light in the parking lot", "type": "global_context",
    "text": "Traffic light on the highway median", "type": "global_context",
    "text": "Traffic light next to the storefront", "type": "global_context",
    "text": "Traffic light by the bus stop", "type": "background_relation",
    "text": "Traffic light near the pedestrian crossing sign", "type": "background_relation",
    "text": "Traffic light beside the kiosk", "type": "background_relation",
    "text": "Stop sign above the crosswalk", "type": "object_swap",
    "text": "Street lamp above the crosswalk", "type": "object_swap",
    "text": "Billboard above the crosswalk", "type": "object_swap"
  ]
}
```

Figure 5: The prompt template used for constructing our SCaR benchmark with GPT-4V where the text in red varies for each sample.

**Prompt Template for LLM-based Filtering**

You are a highly-capable AI assistant designed for meticulous data verification. Your task is to analyze a single data sample from a dataset generation pipeline. This sample consists of a "ground-truth" caption and an array of 9 "negatives" captions, each with a corresponding type. You will strictly adhere to the following rules and provide a structured JSON output.

## Input Data:

You will receive a single JSON object containing a ground-truth string and a negatives array.

## Verification Rules:

For each data sample, you must perform the following checks:

1. Ground-Truth Caption Verification:
   - Structure Check: The ground-truth caption must be verifiable as having a three-part structure: `<object> <relation> <scene>`.
     - `<object>`: The main noun or object of the caption.
     - `<relation>`: A verb or prepositional phrase describing the object's action or its position relative to the scene or other objects.
     - `<scene>`: The broader context or location where the object and relation take place.
   - Example: For "Traffic light above the crosswalk", the parts are:
     - `<object>`: "Traffic light"
     - `<relation>`: "above"
     - `<scene>`: "the crosswalk"
   - Output: If the structure is correct, extract these three elements. If not, mark the sample as "failed".

2. Negative Captions Verification:
   - Count Check: There must be exactly 9 negatives in the negatives array.
   - Type Check: The 9 negatives must be split exactly as 3 'global context', 3 'background relation', and 3 'object swap'. No other types are allowed.
   - Redundancy Check: Within each negative type, the captions must be unique. No two 'global context' captions can be identical, no two 'background relation' captions can be identical, and no two 'object swap' captions can be identical.

3. Cross-Sample Verification (Crucial for filtering):
   - Global Context Negative Check:
     - Each 'global context' negative must have the exact same `<object>` and `<relation>` as the ground-truth caption.
     - The `<scene>` of the 'global context' negative must be semantically distinct and not a synonym or near-synonym of the ground-truth `<scene>`. For example, "in the kitchen" is distinct from "on the beach", but "zoo" is not distinct from "safari park".
   - Background Relation Negative Check:
     - Each 'background relation' negative must have the exact same `<object>` and `<scene>` as the ground-truth caption.
     - The `<relation>` of the 'background relation' negative must be semantically distinct and not a synonym of the ground-truth `<relation>`.
   - Object Swap Negative Check:
     - Each 'object swap' negative must have the exact same `<relation>` and `<scene>` as the ground-truth caption.
     - The `<object>` of the 'object swap' negative must be a different, distinct object class from the ground-truth `<object>`. It must not be a synonym, hyponym, or hypernym. For example, "Traffic light" and "Stoplight" are invalid swaps. "Bowl" and "Mug" are invalid swaps. "Cat" and "Dog" are invalid swaps. The new object must be contextually plausible.

## Output Format:

You must provide a single JSON object as your output. Do not include any Markdown, code fences, or additional commentary.

The JSON object should have the following keys:

"status": A string, either "passed" if all checks succeed, or "failed" if any check fails.

"reasons": An array of strings. If the status is "failed", this array should contain a detailed list of every rule that was violated (e.g., "ground-truth caption lacks a scene", "Object swap negative 1 is a synonym of the ground-truth object", "Number of global context negatives is not 3"). If the status is "passed", this array should be empty.

"ground_truth_elements": An object containing the extracted components of the ground-truth caption. This should be populated only if the ground-truth verification passes.

"object": The extracted object string.

"relation": The extracted relation string.

"scene": The extracted scene string.

"negative_elements": An array of objects. Each object corresponds to a negative caption and should contain:

"text": The original negative caption text.

"type": The original negative caption type.

"object": The extracted object string from the negative caption.

"relation": The extracted relation string from the negative caption.

"scene": The extracted scene string from the negative caption.

Figure 6: The prompt template used for verifying the collected samples via LLM-based filtering.

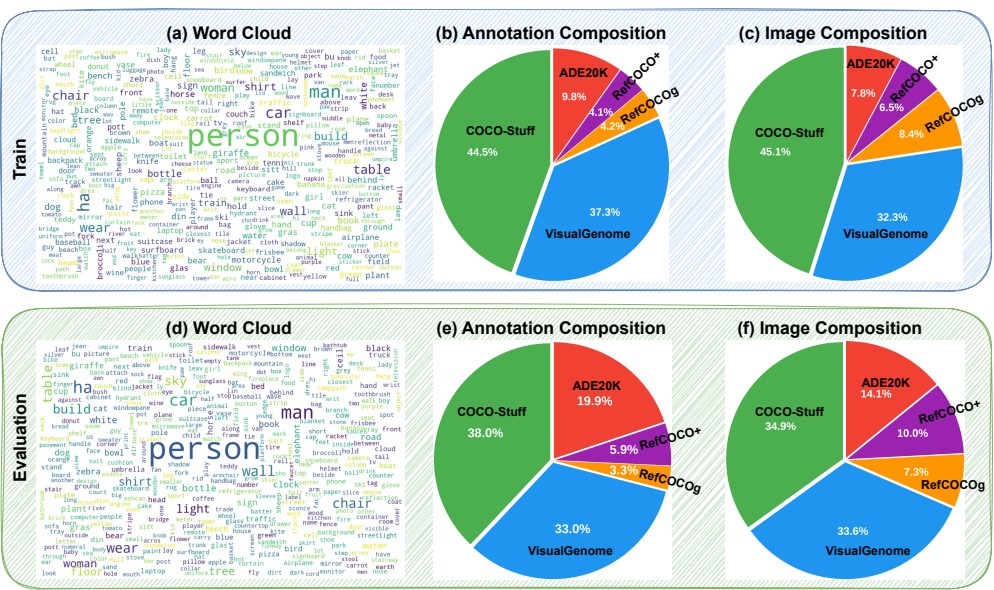

Figure 7: Detailed statistics of the SCaR train and evaluation sets. (a), (d) Word clouds for the candidates. (b), (e): Dataset compositions in terms of numbers of samples. (c), (f): Dataset compositions in terms of numbers of images.

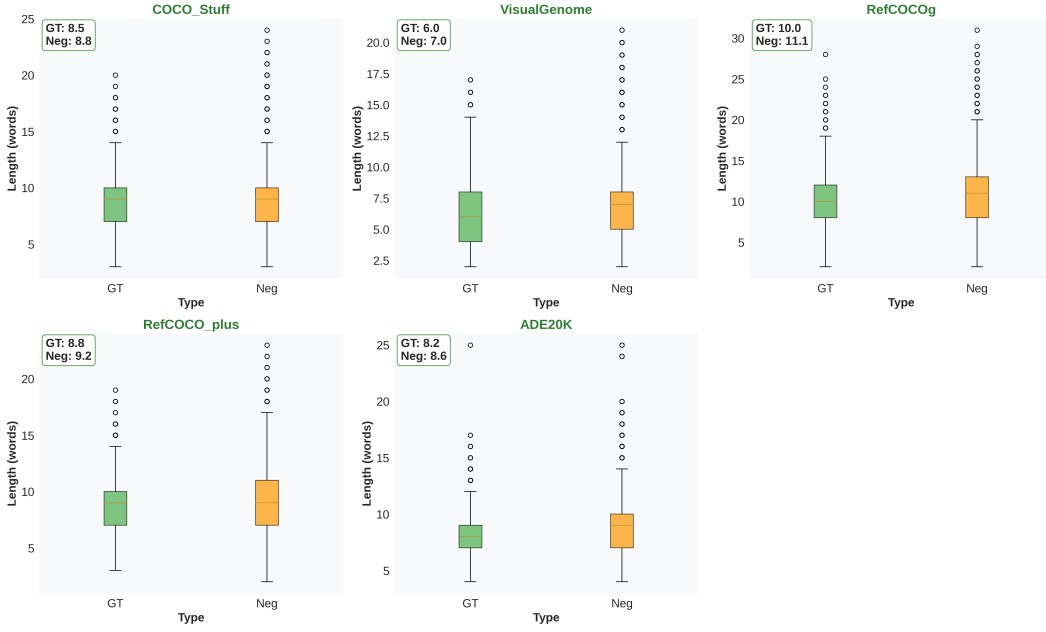

Figure 8: Distributions of sentence lengths for ground-truth and negative captions across each dataset.

Table 8: Examples of datasets in SCaR. The instruction across all datasets is *Find the caption that best describes the segmented object, considering both local details and global context in the given image. Referring object bbox: {bbox}.* Each first candidate in red is the ground-truth caption.

| Dataset | Query Image with bbox | Candidates |
|---|---|---|
| RefCOCO+ (Yu et al., 2016) | | Motorcycle in forefront fully shown.
Motorcycle in the garage fully shown.
Motorcycle on a racetrack fully shown.
Motorcycle in a field fully shown.
Motorcycle with a helmet placed on the seat in forefront.
Motorcycle being washed in forefront.
Motorcycle loaded with packages in forefront.
Bicycle in forefront fully shown.
Scooter in forefront fully shown.
Horse in forefront fully shown. |
| RefCOCOg (Mao et al., 2016) | | The bench closest to the palm tree and on a concrete pedestal at the beach.
The bench closest to the palm tree and on a concrete pedestal in a city bus station.
The bench closest to the palm tree and on a concrete pedestal in a shopping mall atrium.
The bench closest to the palm tree and on a concrete pedestal in a hospital waiting area.
The bench with a row of flower pots on its seat and on a concrete pedestal at the beach.
The bench covered in colorful graffiti and on a concrete pedestal at the beach.
The bench holding a stack of books and on a concrete pedestal at the beach.
The playground slide closest to the palm tree and on a concrete pedestal at the beach.
The trash can closest to the palm tree and on a concrete pedestal at the beach.
The bicycle rack closest to the palm tree and on a concrete pedestal at the beach. |
| VisualGenome (Krishna et al., 2017) | | Mouth of sculpture by the waterfront.
Mouth of sculpture in a museum gallery.
Mouth of sculpture in a lush garden.
Mouth of sculpture on a mountaintop.
Mouth of sculpture blowing smoke by the waterfront.
Mouth of sculpture illuminated by spotlights by the waterfront.
Mouth of sculpture eating an apple by the waterfront.
Fin of sculpture by the waterfront.
Ear of sculpture by the waterfront.
Tail of sculpture by the waterfront. |
| COCO-Stuff (Caesar et al., 2018) | | Cat sitting in front of a computer screen.
Cat sitting on a kitchen countertop.
Cat sitting in a garden.
Cat sitting on a window sill.
Cat pawing at a coffee cup in front of a computer screen.
Cat curled up sleeping in front of a computer screen.
Cat playing with headphones in front of a computer screen.
Rabbit sitting in front of a computer screen.
Dog sitting in front of a computer screen.
Parrot sitting in front of a computer screen. |
| ADE20K (Zhou et al., 2017) | | Fan standing near the chairs in a glass-roofed lounge.
Fan standing near the chairs on a subway platform.
Fan standing near the chairs in a hospital waiting area.
Fan standing near the chairs in a gymnasium.
Fan blowing onto a group of potted plants in a glass-roofed lounge.
Fan hanging from the ceiling above the chairs in a glass-roofed lounge.
Fan surrounded by scattered magazines on the floor in a glass-roofed lounge.
Sculpture standing near the chairs in a glass-roofed lounge.
Lamp standing near the chairs in a glass-roofed lounge.
Plant standing near the chairs in a glass-roofed lounge. |

collection process described in Sec. 3.2, we observe that the word frequency distributions of ground-truth and negative captions are closely aligned across splits, suggesting that the benchmark is not dominated by a small set of frequent words. Moreover, both image- and annotation-level distributions reveal that COCO-Stuff and VisualGenome contribute the largest proportions of samples, reflecting their broad coverage of scenes and object relationships. Fig. 8 illustrates the sentence length distributions of ground-truth and negative captions across the five datasets in SCaR. Overall, the two distributions are closely aligned, indicating that the synthesized negative captions generated by GPT-4V preserve similar linguistic complexity to the ground-truth annotations rather than introducing trivial artifacts. We also observe slightly heavier tails in the negative caption distributions, suggesting greater variability in sentence length. This property enhances the linguistic diversity within SCaR while maintaining a comparable level of difficulty, thereby preventing models from exploiting superficial cues such as caption length.

## D.5 SCaR Examples

Tab. 8 presents SCaR examples for each dataset, where it can be seen that the five datasets cover diverse domains. Additionally, the examples indicate that naively cropping with bounding boxes would sacrifice global scene context (e.g., "at the beach" in RefCOCOg, "by the waterfront" in VisualGenome, and "in a glass-roofed lounge" in ADE20K).

Table 9: Ablation study with VIRTUE-2B in terms of 1) VLM, 2) Task-specific instructions, 3) Image resolution, 4) Segmentation streamline, 5) Segmentation model choice; 6) # MLPs in the segmentation-language connector; 7) Length of segmentation embeddings; 8) Cross-modal alternatives; and 9) Finetuning SAM-2. The highlighted row denotes the configurations for VIRTUE-2B and VIRTUE-7B.

| 1) Choice | MMEB | SCaR |
| --- | --- | --- |
| Phi-3.5-V | 61.7 | 26.5 |
| Qwen2-VL-2B | 64.8 | 30.4 |

| 2) Instruction | MMEB | SCaR |
| --- | --- | --- |
| ✗ | 59.2 | 24.3 |
| ✓ | 64.8 | 30.4 |

| 3) Resolution | MMEB | SCaR |
| --- | --- | --- |
| 672x672 | 60.1 | 27.6 |
| 1344x1344 | 64.8 | 30.4 |

| 4) Alternative | MMEB | SCaR |
| --- | --- | --- |
| Prompter | 61.0 | 22.7 |
| Cropped | 63.3 | 25.9 |
| Segmentation | 64.8 | 30.4 |

| 5) Choice | MMEB | SCaR |
| --- | --- | --- |
| SAM2.1-S | 60.1 | 21.6 |
| SAM2.1-B+ | 64.8 | 30.4 |
| SAM2.1-L | 63.8 | 27.9 |

| 6) # MLP | MMEB | SCaR |
| --- | --- | --- |
| 1 | 60.4 | 18.6 |
| 2 | 64.8 | 30.4 |
| 3 | 63.5 | 29.7 |

| 7) $|S|$ | MMEB | SCaR |
| --- | --- | --- |
| 64 | 61.1 | 29.9 |
| 256 | 64.8 | 30.4 |
| 1024 | 60.2 | 35.1 |

| 8) Alternative | MMEB | SCaR |
| --- | --- | --- |
| Cross-attention | 63.1 | 30.0 |
| Attention-pool | 55.9 | 28.1 |
| Conv-MLP | 64.8 | 30.4 |

| 9) Finetune | MMEB | SCaR |
| --- | --- | --- |
| ✗ | 64.8 | 30.4 |
| ✓ | 63.5 | 25.8 |

# E  ADDITIONAL EXPERIMENTS

## E.1  ABLATION STUDY

We conduct seven ablation variants to investigate the relative contributions of different components: 1) VLM backbones, 2) with and without an instruction in the query, 3) input image resolution, 4) segmentation streamline alternatives, 5) different sizes of SAM-2, 6) different numbers of MLPs in the segmentation-language connector (Eq. 1), where one MLP projects from $d_s$ to $d$, and three MLPs project from $d_s$ to 768, then to 1024, and finally to $d$; 7) segmentation embedding lengths $|S|$; 8) cross-modal modeling alternatives; and 9) finetuning and freezing SAM-2. For 4) segmentation streamline alternatives, we replace our segmentation model with Prompter from CLOC[5] (Chen et al., 2025), which encodes bounding boxes via a randomly initialized Transformer, or directly crop images for the vision encoder (denoted as Cropped). Both variants concatenate embeddings from nine random crops, similar to the setting of nine sampled points in SAM2.1 for the segmentation-language connector for MMEB. For 8) cross-modal modeling alternatives, we adopt two variants for the segmentation-language connect: Cross-attention, which uses the same architecture as the vision-language connector in Qwen2-VL. Attention-pooling: A natural approach using $k$ learnable queries to pool information from the region features (we used $k = 64$).

As shown in Tab. 9, although Phi-3.5-V has more parameters than Qwen2-VL-2B, the latter yields better results on MMEB and SCaR. Removing text instructions and using lower resolutions lead to a significantly inferior performance. The degraded results from Prompter and cropping suggest that pre-trained SAM2.1 provides not only more precise segmentations than simple crops but also a stronger understanding from its pre-trained knowledge. Although SAM2.1-L performs on par with SAM2.1-B+ in terms of MMEB, its performance degrades on SCaR. In contrast, using SAM2.1-S deleteriously impacts both benchmarks. Additionally, increasing MLP layers and segmentation embedding length $|S|$ does not consistently yield gains across MMEB and SCaR. While increasing segmentation embeddings to 1024 delivers better results on SCaR, we stick to selecting 256 as our configuration to balance computation and effectiveness. Using attention-based cross-modal methods does not yield better results, although cross-attention achieves performance comparable to our convolutional MLPs. Similarly, finetuning SAM-2 performs worse than keeping it frozen–especially on SCaR–since MMEB-train does not contain user-specific visual prompts (i.e., using uniform points). This mismatch biases the segmentation pipeline toward non–visual-interactive scenarios.

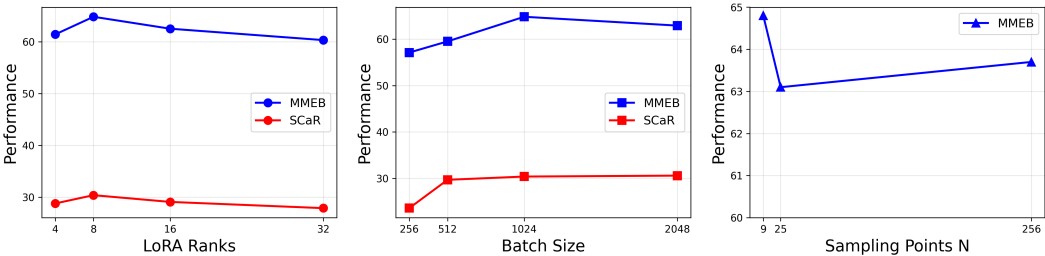

Figure 9: The impacts of varying LoRA ranks, batch sizes, visual-prompt design choice, and sampling points $N$ on VIRTUE-2B.

Table 10: MMEB reevaluation results after finetuning on the SCaR-train dataset.

| | **Per Meta-Task Score** | | | | **Average Score** | | |
|---|---|---|---|---|---|---|---|
| | Classification | VQA | Retrieval | Grounding | IND | OOD | Overall |
| **# Datasets** → | 10 | 10 | 12 | 4 | 20 | 16 | 36 |
| VLM2Vec-2B | 58.7 | 49.3 | 65.0 | 72.9 | 64.9 | 53.3 | 59.7 |
| +SCaR-train | 35.8 | 7.5 | 14.7 | 25.8 | 19.4 | 19.2 | 19.8 |
| VLM2Vec-7B | 62.7 | 56.9 | 69.4 | 82.2 | 71.4 | 58.1 | 65.5 |
| +SCaR-train | 47.5 | 31.8 | 48.6 | 46.3 | 44.4 | 42.0 | 43.4 |
| MMRet-7B | 56.0 | 57.4 | 69.9 | 83.6 | 68.0 | 59.1 | 64.1 |
| +SCaR-train | 45.2 | 38.1 | 51.7 | 77.9 | 51.8 | 46.5 | 49.4 |
| UniME-7B | 60.6 | 52.9 | 67.9 | 85.1 | 68.4 | 57.9 | 66.6 |
| +SCaR-train | 55.0 | 46.4 | 64.3 | 84.5 | 62.7 | 54.3 | 59.0 |
| VIRTUE-2B (Ours) | 64.1 | 55.7 | 68.4 | 78.7 | 69.7 | 58.8 | 64.8 |
| +SCaR-train | 64.7 | 55.1 | 68.3 | 79.2 | 69.8 | 58.7 | 64.9 |
| VIRTUE-7B (Ours) | 65.6 | 60.4 | 71.8 | 87.3 | 74.4 | 61.4 | 68.6 |
| +SCaR-train | 64.4 | 58.1 | 67.0 | 86.4 | 72.2 | 60.0 | 66.8 |

## E.2 HYPERPARAMETER STUDY

We further analyze the impacts of 1) LoRA rank (4, 8, 16, 32), 2) batch size (256, 512, 1024, 2048), and 3) sampling points $N$ used in non-visual-interactive tasks (MMEB). Fig. 9 shows that varying LoRA ranks produces consistent outcomes, with rank 8 showing a slight edge. Furthermore, larger batch sizes improve generalizability for in-batch negatives, with a batch size of 1024 obtaining the best results. We also find that increasing the number of sampling points does not enhance model effectiveness for non-visual-interactive tasks, presumably because key entities within an image for general tasks can be captured with only a few points.

## E.3 REEVALUATING MMEB WITH FURTHER FINETUNING ON SCAR-TRAIN

To evaluate the impact of incorporating SCaR-train on MMEB performance, we reevaluate VLM2Vec-2B, VLM2Vec-7B, MMRet-7B, UniME-7B, VIRTUE-2B, and VIRTUE-7B with SCaR-train (+SCaR-train) (Sec. 5.3) and compare them against the original checkpoints trained solely on MMEB-train. As shown in Tab. 10, all baselines exhibit a degraded performance, likely because the models overfit or shift focus toward the SCaR distribution for incremental learning (Li & Hoiem, 2016). Notably, VLM2Vec and MMRet suffer severe degradation, suggesting stronger susceptibility to catastrophic forgetting (French, 1999) when their training rely more heavily on additional data. In contrast, UniME, with its multi-stage contrastive learning design, demonstrates greater robustness against forgetting. Meanwhile, VIRTUE maintains a performance comparable to the original MMEB-trained checkpoints, with VIRTUE-2B even achieving a slight 0.1-point gain. We attribute

---

[5]CLOC cannot be directly evaluated or reproduced, as its model, code, and parts of training data are not publicly available. We attempted to contact the authors but did not receive a response.

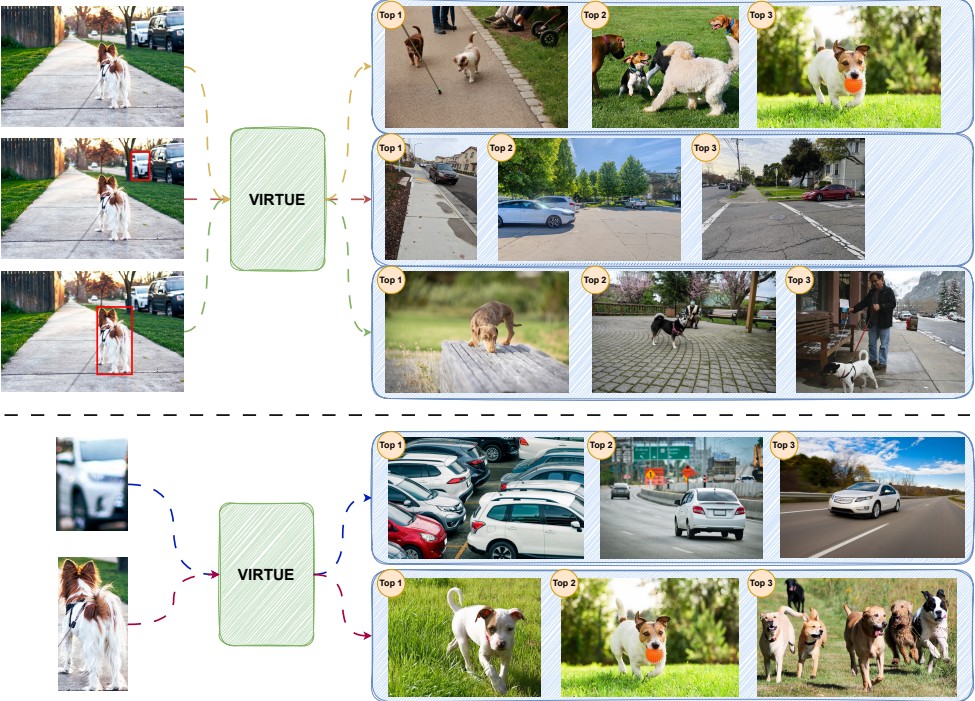

Figure 10: In-the-wild visual-interactive (top) and naive cropping (bottom) image-to-image retrieval scenarios. VIRTUE-7B leverages visual prompts (bounding boxes in this paradigm) to guide the retrieval of regions of interest while accounting for both entity-level details and global scene context.

this stability to the segmentation streamline, which enables the use of optional visual prompts and makes VIRTUE more adaptable as a universal embedder capable of handling both visual-interactive and non-visual-interactive tasks.

### E.4 CASE STUDIES FOR VIRTUE'S VISUAL-INTERACTIVE CAPABILITIES

With visual interaction, VIRTUE enables new applications such as segment-level retrieval, where users select a region of interest to fetch semantically matching images, and entity-level hinting for on-the-fly correction, thereby extending the utility of embedding-based systems far beyond traditional global matching.

### E.4.1 VISUAL-INTERACTIVE IN-THE-WILD I2I RETRIEVAL

While evaluating VIRTUE on SCaR confirms its ability to interact with visual interactions, we further conduct in-the-wild visual-interactive image-to-image retrieval. Similar to the procedure we constructed SCaR, we prompt GPT-4o with "Generate some text captions that need to consist of "object, relation, scene" for searching images. Only contain 2 entities within an image is sufficient.". Then, we use the generated prompt to the Google Search API[6] to get six *ground-truth* images (three depicting cars parked on the road and three depicting dogs on the sidewalk with nearby grass), and subsequently instruct GPT-4o to generate 20 negative captions conditioned on the selected text caption. Afterwards, those negative captions are used to search the corresponding images as negative candidates.

Fig. 10 illustrates the visual-interactive (top three rows) and naive cropping (bottom two rows) image-to-image paradigm enabled by VIRTUE-7B. The first row depicts conventional retrieval applications, also supported by VIRTUE, where images are retrieved based on holistic query image information. What distinguishes VIRTUE is its ability to incorporate user-specified visual prompts,

---

[6]https://developers.google.com/custom-search/

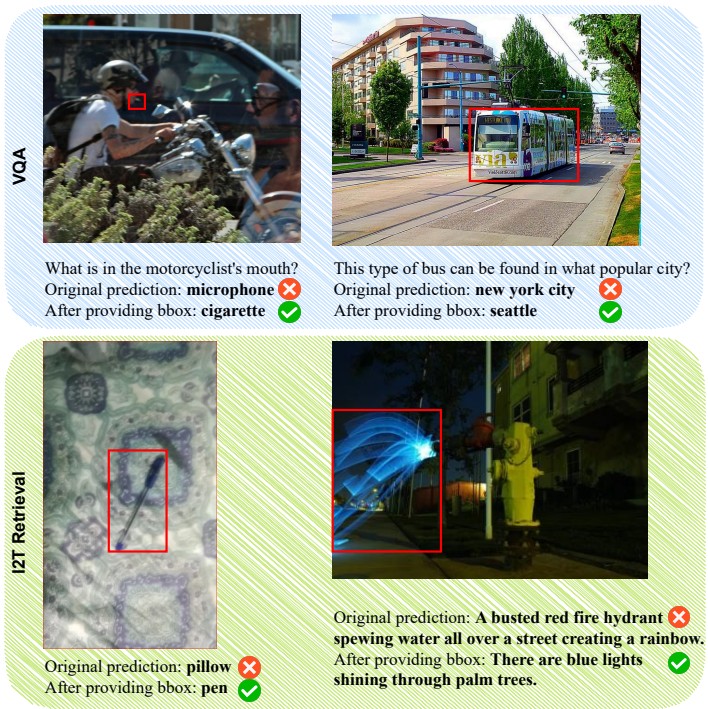

Figure 11: On-the-fly correction with explicitly bounding box hinting of VQA and I2T retrieval paradigms. All of them are used with VIRTUE-2B.

as demonstrated in the second and third rows within the figure. For instance, selecting the parking car directs VIRTUE to focus on the car while preserving the surrounding scene context (e.g., grass or trees along the road). Similarly, selecting the dog on a sidewalk guides the model to retrieve dogs on sidewalks with nearby grass or trees. Attributed to the segmentation streamline in VIRTUE, both scenarios illustrate novel possibilities for interactive querying between humans and embedding models. On the flip side, naive cropping (bottom two rows) discards scene context, leading the model to retrieve the selected entity in mismatched contexts from the given image (e.g., a car running on the street or a dog running on the grass in the retrieved results, even though neither is moving in the query image).

### E.4.2 ON-THE-FLY CORRECTION WITH VISUAL HINTING

To delve into the capability of on-the-fly correction, we randomly sample MMEB cases that VIRTUE-2B initially misclassifies. We then manually provide the correct regions as visual prompts and re-run inference. As illustrated in Fig. 11, VIRTUE successfully corrects predictions not only for VQA but also for retrieval tasks, relying solely on visual hints at inference time without additional finetuning. This demonstrates a new mode of applicability for VLM-based embedding models, where users can guide the model interactively while avoiding the computational cost of conventional finetuning. Although direct cropping of hinted regions may also yield corrections, visual prompting in VIRTUE eliminates the need for heuristic preprocessing and proves more robust in challenging cases. For example, in VQA tasks, cropping can fail when the region excludes crucial contextual information (e.g., the motorcyclist), whereas visual prompts enable the model to integrate both global and fine-grained cues.

### E.5 QUALITATIVE RESULTS ON SCAR

Fig. 12 presents qualitative comparisons between baselines and our VIRTUE-7B model trained with SCaR-train. We observe that the baselines frequently misinterpret relational cues (e.g., "player with head down" to "player picking up a bat") and even retrieve entirely incorrect objects (e.g., "tree" to "cow and statue"). A likely reason is that their representations are dominated by LLM-

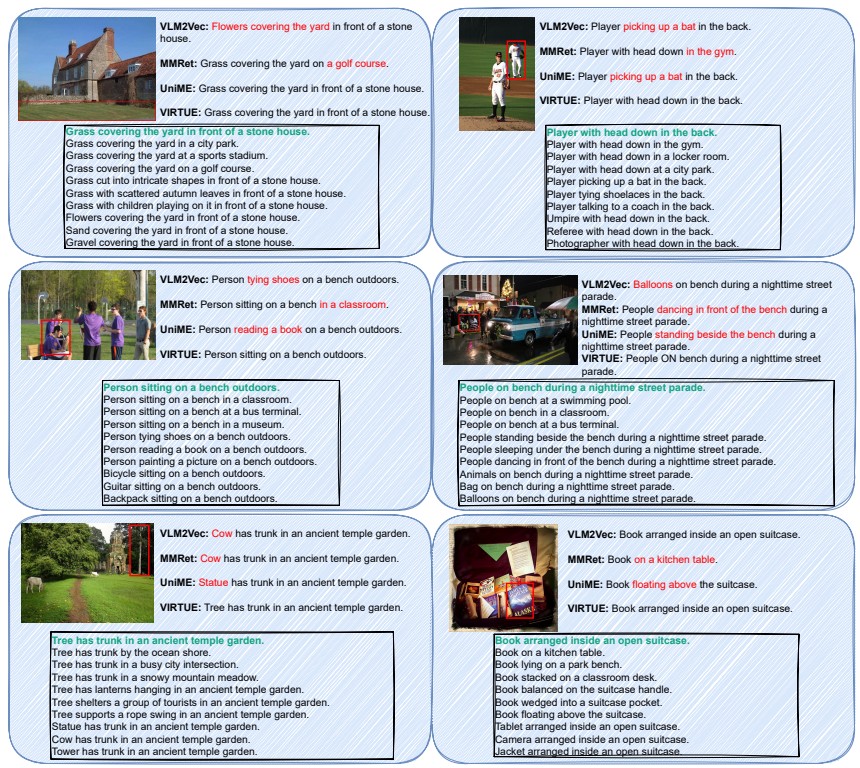

Figure 12: Qualitative comparison of VLM2Vec-7B, MMRet-7B, UniME-7B, and VIRTUE-7B. Ground-truth captions are shown in green and incorrect ones in red.

derived features (Tong et al., 2024), which tend to overlook spatial information such as bounding boxes in the text query. In contrast, VIRTUE incorporates a segmentation-based streamline that augments regions of interest with fine-grained visual representations, effectively guiding retrieval toward semantically and spatially accurate matches.

E.6 ANALYSIS ON LATENCY AND MEMORY

We report the average end-to-end inference time (ms) and GPU memory (GB) for four distinct operational modes of VIRTUE-7B on the SCaR benchmark (I2T) and provide baselines (VLM2Vec, MMRet, and UniME) for I2I (retrieval and visual grounding) performance from MMEB, as shown

Table 11: Analysis on inference time and memory for interactive I2T (SCaR) and I2I (retrieval and visual grounding of MMEB) of different query modes. For VIRTUE-7B, query modes with cropping and text-only remove the segmentation streamline.

|  | Query Mode | Time (per sample) | Memory | Performance |
|---|---|---|---|---|
| **I2T (SCaR)** | | | | |
| VIRTUE-7B | w/ visual prompts | 422ms | 18.8GB | 27.8 |
|  | w/o visual prompts (uniform points) | 427ms | 18.8GB | 16.1 |
|  | Cropping | 139ms | 17.7GB | 22.8 |
|  | Text-only | 340ms | 17.7GB | 23.9 |
| **I2I (Retrieval and Visual Grounding on MMEB)** | | | | |
| VIRTUE-7B | w/ uniform points | 402ms | 18.8GB | 70.7 |
|  | w/o uniform points | 357ms | 17.7GB | 46.5 |
| VLM2Vec-7B | N/A | 406ms | 18.6GB | 67.9 |
| MMRet-7B | N/A | 538ms | 29.0GB | 75.8 |
| UniME-7B | N/A | 419ms | 28.9GB | 66.4 |

in Tab. 11. For cropping and text-only modes, we remove the visual prompt streamline. All I2T evaluations are reported at matched precision (bf16), and all results are obtained with a batch size of 1 on a single H100 GPU. While cropping and text-only modes offer better latency and memory, the performance becomes significant inferior. We further include the removal of visual prompts for I2I scenarios, which is more efficient but degrading significantly. Moreover, using uniform points is more efficient than the existing baselines.

## E.7   DETAILED SCORES OF MMEB

Tab. 12 presents detailed results of 6 VLM-based models that are trained with MMEB for comprehensive comparisons. The performances are sourced from the corresponding papers. Due to the space limits, the detailed scores of CLIP, BLIP2, SigLIP, OpenCLIP, UniIR, Magiclens, E5-V, GME, and LamRA can be referred from their official papers.

Table 12: Detailed results of the VLM-based baselines and our VIRTUE on MMEB. The 16 out-of-distribution datasets are highlighted in pink.

| | VLM2Vec-2B | **VIRTUE-2B** | MMRet-7B | VLM2Vec-7B | UniME-7B | **VIRTUE-7B** |
|---|---|---|---|---|---|---|
| **Classification (10 tasks)** | | | | | | |
| ImageNet-1K | 77.5 | 80.1 | 58.1 | 80.1 | 71.3 | 82.3 |
| N24News | 73.7 | 78.4 | 71.3 | 79.7 | 79.5 | 81.1 |
| HatefulMemes | 58.3 | 67.5 | 53.7 | 69.7 | 64.6 | 74.1 |
| VOC2007 | 74.3 | 83.1 | 85.0 | 80.7 | 90.4 | 85.4 |
| SUN397 | 73.8 | 74.3 | 70.0 | 77.4 | 75.9 | 78.3 |
| Place365 | 35.3 | 36.4 | 43.0 | 37.4 | 45.6 | 39.6 |
| ImageNet-A | 50.9 | 53.4 | 36.1 | 58.1 | 45.5 | 55.2 |
| ImageNet-R | 84.7 | 88.3 | 71.6 | 73.9 | 78.4 | 82.5 |
| ObjectNet | 37.1 | 48.9 | 55.8 | 40.1 | 36.4 | 44.3 |
| Country-211 | 21.5 | 30.5 | 14.7 | 29.8 | 18.7 | 32.8 |
| **Average** | 58.7 | 64.1 | 56.0 | 62.7 | 60.6 | 65.6 |
| **VQA (10 tasks)** | | | | | | |
| OK-VQA | 48.5 | 56.7 | 73.3 | 56.8 | 68.3 | 59.6 |
| A-OKVQA | 39.5 | 50.5 | 56.7 | 47.3 | 58.7 | 52.2 |
| DocVQA | 82.5 | 87.9 | 78.5 | 89.7 | 67.6 | 91.7 |
| InfographicsVQA | 47.7 | 53.4 | 39.3 | 60.0 | 37.0 | 63.7 |
| ChartQA | 42.3 | 48.2 | 41.7 | 56.9 | 33.4 | 61.0 |
| Visual7W | 51.2 | 52.3 | 49.5 | 52.7 | 51.7 | 55.0 |
| ScienceQA | 30.7 | 40.9 | 45.2 | 38.5 | 40.5 | 46.7 |
| VizWiz | 38.6 | 44.3 | 51.7 | 39.9 | 42.7 | 43.3 |
| GQA | 48.3 | 48.7 | 59.0 | 55.1 | 63.6 | 54.7 |
| TextVQA | 63.3 | 74.1 | 79.0 | 71.6 | 65.2 | 76.4 |
| **Average** | 49.3 | 55.7 | 57.4 | 56.9 | 52.9 | 60.4 |
| **Retrieval (12 tasks)** | | | | | | |
| VisDial | 74.3 | 78.5 | 83.0 | 81.9 | 79.7 | 83.3 |
| CIRR | 46.8 | 56.0 | 61.4 | 51.1 | 52.2 | 60.4 |
| VisualNews_t2i | 73.1 | 75.1 | 74.2 | 80.5 | 74.8 | 80.8 |
| VisualNews_i2t | 73.7 | 77.9 | 78.1 | 81.2 | 78.8 | 82.5 |
| MSCOCO_t2i | 73.4 | 73.9 | 78.6 | 77.2 | 74.9 | 78.0 |
| MSCOCO_i2t | 68.5 | 72.9 | 72.4 | 73.9 | 73.8 | 76.3 |
| NIGHTS | 66.3 | 66.9 | 68.3 | 67.6 | 66.2 | 70.6 |
| WebQA | 85.9 | 88.7 | 90.2 | 88.3 | 89.8 | 91.2 |
| FashionIQ | 14.0 | 15.0 | 54.9 | 17.1 | 16.5 | 18.1 |
| Wiki-SS-NQ | 54.2 | 60.8 | 24.9 | 62.3 | 66.6 | 66.3 |
| OVEN | 68.3 | 69.2 | 87.5 | 66.5 | 55.7 | 67.2 |
| EDIS | 81.2 | 85.6 | 65.6 | 85.7 | 86.2 | 86.6 |
| **Average** | 65.0 | 68.4 | 69.9 | 69.4 | 67.9 | 71.8 |
| **Visual Grounding (4 tasks)** | | | | | | |
| MSCOCO | 66.5 | 70.7 | 76.8 | 76.5 | 76.5 | 80.5 |
| RefCOCO | 80.9 | 86.0 | 89.8 | 89.3 | 89.3 | 94.2 |
| RefCOCO-matching | 75.7 | 83.1 | 90.6 | 90.6 | 90.6 | 92.0 |
| Visual7W-pointing | 68.3 | 75.0 | 77.0 | 84.1 | 84.1 | 82.6 |
| **Average** | 72.9 | 78.7 | 83.6 | 82.2 | 85.1 | 87.3 |
| **Final Score (36 tasks)** | | | | | | |
| **IND** | 64.9 | 69.7 | 68.0 | 71.4 | 68.4 | 74.4 |
| **OOD** | 53.3 | 58.8 | 59.1 | 58.1 | 57.9 | 61.4 |
| **Average** | 59.7 | 64.8 | 64.1 | 65.5 | 66.6 | 68.6 |

