# OpenReview forum: "VIRTUE: Visual-Interactive Text-Image Universal Embedder"
_ICLR.cc/2026/Conference — ICLR 2026 Poster_

### Official Review · Reviewer_jcPk · 2025-10-27

**Soundness:** 3
**Presentation:** 3
**Contribution:** 3
**Rating:** 6
**Confidence:** 4

**Summary:**

This paper presents a new visual-interactive text-image universal embedding model designed to address the limitations of existing embedding models in handling visual cues such as bounding boxes, masks, and clicks. It combines the segmentation model (SAM-2) with a pre-trained vision-language model (VLM), enabling users to visually specify regions of interest within images, thereby enhancing the model's understanding of entity-level information in images.To evaluate the visual interaction capabilities of VIRTUE, the authors also constructed a large-scale benchmark named SCaR.

**Strengths:**

1. New Task: This paper introduces a novel visual-text embedding paradigm that, for the first time, incorporates visual cues such as bounding boxes, points, and masks into a universal embedding model. This approach overcomes the limitations of traditional models that exclusively support textual prompts and expands the possibilities for human-computer interaction.
2. Valuable Benchmarks: SCaR is the first large-scale benchmark designed for visual-interactive image-text retrieval, filling a gap in this field with significant diversity and challenges.
3. Clear writing and comprehensive experiments.

**Weaknesses:**

The primary contribution of this paper lies in the introduction of a new dataset. The proposed method does not present any technical innovations; rather, it merely combines SAM and VLM. Consequently, the technical contributions of this paper are quite limited.

**Questions:**

1. VIRTUE has brought significant performance improvements. To clarify the respective contributions of the dataset and the methodology, I would like to konw performance variations of other methods in the two scenarios of using SCaR and not using SCaR, as well as the outcomes of the proposed method when SCaR is not employed.
2. The generation of negative samples in the SCaR dataset relies on GPT-4V. Are there any biases or issues related to semantic repetition?

---

> ### Author Response · Authors · 2025-11-20
> **Response to Reviewer jcPk**
>
> We sincerely appreciate your positive feedback. Our point-to-point responses to your comments are listed below.
>
> > **[W1] Limited technical innovation.**
>
> We would like to clarify that combining SAM and a VLM into a unified embedding model is non-trivial and introduces several architectural and algorithmic challenges, such as how to design the segmentation-language connector, how to select an appropriate segmentation encoder, how different sizes of SAM affect performance, and how many segmentation tokens should be used. These design questions and their analyses are discussed in **Appendix E.1**.
>
> To further emphasize our technical contributions, we have conducted additional studies that provide clear guidelines and best practices for integrating new visual prompt modalities into embedding models, including (1) cross-token interactions (**Table 9-(8)**), (2) freezing and unfreezing SAM-2 (**Table 9-(9)**), (3) whether VIRTUE actually uses visual prompts (**Section 5.4 with Table 4**), (4) robustness to visual prompts (**Section 5.4 with Figure 4**), and (5) latency and memory use (**Appendix E.6 with Table 11**).
>
> We have added a statement summarizing these systematic design guidelines in the revised manuscript (**L110-111**). These analyses provide the architectural principles and practical blueprints necessary for enabling visual-interactive capabilities within an MLLM embedding space, which we believe constitute meaningful technical contributions.
>
> > **[Q1] Relative performance contributions between the dataset and the method.**
>
> As shown in Tables 3 (with and without SCaR-train for SCaR) and 7 (with and without SCaR-train for MMEB), we already include the performance variations for both baselines (i.e., VLM2Vec-2B, VLM2Vec-7B, MMRet-7B, UniME-7B) and our proposed VIRTUE-2B as well as VIRTUE-7B, both with and without training on SCaR-train.
>
> For clarity, we summarize them below. without SCaR-train (i.e., x) refers to only training on MMEB, and with SCaR-train (i.e., v) refers to finetuning on SCaR-train from the MMEB trained checkpoint (L1231-1233). The table demonstrates the positive contributions of our method (e.g., VIRTUE-2B (x) vs. VLM2Vec-2B (x)) as well as relevant improvements from using SCaR (e.g., VIRTUE-2B (x) vs. VIRTUE-2B (v)).
>
> | | Benchmark -> | MMEB | SCaR |
> | --- | :---: | --- | --- |
> | | Use SCaR-train? | | |
> | VLM2Vec-2B | x | 59.7 | 24.1 |
> | VLM2Vec-2B | v | 19.8 | 46.7 |
> | VLM2Vec-7B | x | 65.5 | 22.9 |
> | VLM2Vec-7B | v | 43.4 | 34.5 |
> | MMRet-7B     | x | 64.1 | 22.5 |
> | MMRet-7B     | v | 49.4 | 37.6 |
> | UniME-7B      | x | 66.6 | 26.3 |
> | UniME-7B      | v | 59.0 | 49.4 |
> | VIRTUE-2B (Ours)    | x | **64.8** | **30.4** |
> | VIRTUE-2B (Ours)    | v | **64.9** | **56.2** |
> | VIRTUE-7B (Ours)    | x | **68.6** | **27.8** |
> | VIRTUE-7B (Ours)    | v | **66.8** | **56.9** |
>
> > **[Q2] Are there any biases or issues related to semantic repetition for generating negative samples in SCaR?**
>
> Great question! We did observe that GPT-4V sometimes leaned toward generating negative samples with semantic repetition, specifically by using synonyms or hyponyms/hypernyms (e.g., zoo vs. safari park; traffic light vs. stoplight; girl vs. woman). Additionally, when merely asked to swap entities, it frequently produced commonly used entities (e.g., chairs, tables, cats).
>
> To mitigate these issues, our prompt template (Figure 4) explicitly addresses both observations:
> - It clearly instructs the model to replace entities/scenes with those that are semantically distinct and to avoid using synonyms or hyponyms/hypernyms.
> - We included a "Creativity and Diversity Constraint" to encourage GPT-4V to maximize variety within each negative sample type.
> In addition, we applied WordNet during the LLM-then-human verification stage to detect and eliminate negative elements that were synonyms of the corresponding gold elements. The resulting dataset statistics, shown in the word cloud (Figure 6), confirm this strategy's success by demonstrating greater diversity and significantly reduced bias.

---

> ### Author Response · Authors · 2025-11-26
> **Rebuttal Followup**
>
> Dear reviewer jcPk,
>
> As the discussion period ends in less than a week, we wanted to check in to see whether you have any remaining questions. **We would be happy to clarify further, and grateful for any other feedback you may povide**. We really appreciate your time engaged in the review and rebuttal phase.
>
> Thank you very much and look forward to your replies!
>
> Best regards,
>
> Authors of Paper 4205

---

### Official Review · Reviewer_a385 · 2025-10-27

**Soundness:** 3
**Presentation:** 3
**Contribution:** 2
**Rating:** 6
**Confidence:** 4

**Summary:**

This paper proposes a visual-prompt MLLM-based embedding model and a corresponding new benchmark for measuring visual-prompt retrieval. The authors observe that existing MLLM-based embedding models can only accept textual instructions. Therefore, they proposed a new model VIRTUE that leverages similar visual prompts with SAM. Experiments are conducted on both the proposed benchmark and an existing benchmark.

**Strengths:**

1. The new problem setting is interesting and novel. I think visual prompts are complimentary to textual prompts to MLLM-based embedding models and can be practical in real-world applications.

2. The paper contributes a new method and a new benchmark which may be beneficial for both the methodology and data perspectives. Meanwhile, the method itself seems reasonable.

3. Experiments on both the proposed benchmark and existing benchmark seem to validate the effectiveness of the proposed method.

**Weaknesses:**

1. The scope of the new benchmark seems narrow. The main task of the new benchmark is caption retrieval. However, most existing benchmarks (e.g., MMEB and MBEIR[1]) for these models contain more tasks such as VQA retrieval, composed retrieval, or visual-document retrieval. The proposed benchmark seems to be more narrow than existing benchmarks that can undermine its universality.

2. The technical novelty seems to be limited. The model seems like a combination of existing MLLM-based embedding models and SAM. The textual prompts and visual prompts are from existing works and the forwarding and training process are similar to existing MLLM-based embedding models.

3. Experiments are only conducted on Qwen2-VL models. It is also recommended to validate the method on different herds of MLLMs. Meanwhile, other important baselines GME-QWen[2] which also uses QWen2-VL  are missing.

4. While handling non-visual-interactive scenarios, the uniform visual prompts seem to be redundant. Since all inputs are concatenated into one sequence, I think directly discarding the visual prompts can be more efficient in this case.

1. Wei C, Chen Y, Chen H, et al. Uniir: Training and benchmarking universal multimodal information retrievers[C]//European Conference on Computer Vision. Cham: Springer Nature Switzerland, 2024: 387-404.

2. Zhang X, Zhang Y, Xie W, et al. GME: Improving Universal Multimodal Retrieval by Multimodal LLMs[J]. arXiv preprint arXiv:2412.16855, 2024.

**Questions:**

See above weakness

---

> ### Author Response · Authors · 2025-11-20
> **Response to Reviewer a385**
>
> Thank you for your careful evaluation of our paper and for providing constructive feedback. Our point-to-point responses to your comments are summarized below.
>
> > **[W1] Narrow scope of the new benchmark. The main task is caption retrieval.**
>
> While SCaR's primary format is caption retrieval, we emphasize that it **also functions as a challenging composed and reasoning retrieval task** since the negative candidates are not randomly selected. Instead, they are systematically constructed by **swapping scenes, relations, and objects** (see examples in Figure 2 and Table 8). This design explicitly aims to challenge models' visual reasoning and compositionality concerning a given region of interest. Moreover, SCaR is diverse in its domain coverage, including 1 million samples across five datasets (RefCOCO+, RefCOCOg, VisualGenome, COCO-Stuff, and ADE20k).
>
> The main distinction is the *type* of capability evaluated: Existing benchmarks (e.g., MMEB, MBEIR) cover more tasks but cannot effectively evaluate visual interaction capabilities for embedding models. This limitation is the primary motivation behind SCaR: to provide a visual-interactive benchmark that complements existing general benchmarks.
>
> Finally, to clarify our claim: the *universality* discussed in the paper applies to our proposed VIRTUE models, which demonstrate strong performance on both conventional benchmarks (like MMEB) and our new visual-interactive SCaR benchmark--not to the SCaR benchmark itself.
>
> > **[W2] Limited technical novelty. Combination of existing MLLM and SAM and training process are similar to existing MLLM-based embedding models.**
>
> Thank you for acknowledging that the training process of our method is similar to existing MLLM-based embedding models, a design choice that grants **adaptability** to various MLLMs as shown in Table 9-(1). We would like to clarify that combining SAM and a VLM into a unified embedding model is non-trivial and introduces several architectural and algorithmic challenges, such as how to design the segmentation-language connector, how to select an appropriate segmentation encoder, how different sizes of SAM affect performance, and how many segmentation tokens should be used. These design questions and their analyses are discussed in **Appendix E.1**.
>
> To further emphasize our technical contributions, we have conducted additional studies that provide clear guidelines and best practices for integrating new visual prompt modalities into embedding models, including (1) cross-token interactions (**Table 9-(8)**), (2) freezing and unfreezing SAM-2 (**Table 9-(9)**), (3) whether VIRTUE actually uses visual prompts (**Section 5.4 with Table 4**), (4) robustness to visual prompts (**Section 5.4 with Figure 4**), and (5) latency and memory use (**Appendix E.6 with Table 11**).
>
> We have added a statement summarizing these systematic design guidelines in the revised manuscript (**L110-111**). These analyses provide the architectural principles and practical blueprints necessary for enabling visual-interactive capabilities within an MLLM embedding space, which we believe constitute meaningful technical contributions.
>
> > **[W3] Recommendation to validate methods on different herds of MLLMs as well as GME-Qwen that also uses Qwen2-VL.**
>
> As mentioned in Baselines (L363-366), we would like to clarify that:
> - GME-Qwen Baseline: We already included the GME-Qwen model as a baseline (denoted as GME), which exactly uses Qwen2-VL, with full results presented in **Tables 2 and 3**.
> - Different herds of MLLMs: Our baseline comparisons in **Tables 2 and 3** also already include diverse MLLM families, including Qwen2-VL (for GME, LamRA, VLM2Vec) and LLaVA-1.6 (MMRet, UniME), and LLaVA-NeXT (E5-V) models.
>
> > **[W4] Directly discarding the visual prompts can be more efficient for non-visual-interactive scenarios.**
>
> Thank you for this excellent suggestion! We thoroughly investigated this idea with two variants: 1) Discard visual prompts only during inference (w/o inference); 2) Discard visual prompts during both training and inference (w/o all).
>
> From the below table, discarding the visual prompts during inference degrades significantly due to the input misalignment between training and inference for the LLM to process, although it indeed becomes more efficient (see Appendix E.6 for the latency study). Removing the visual prompts for both training and inference degrades performance more, indicating that the uniform visual prompts are critical because they effectively enrich the model with fine-grained object-level details. We have included this study in **Section 5.4 (Table 5)**. We appreciate again for this feedback.
>
> | | Visual Prompts | MMEB |
> | --- | --- | --- |
> | VIRTUE-7B | Uniform | **68.6** |
> | | w/o inference | 24.1 |
> | | w/o all | 65.5 |
> | VIRTUE-7B-SCaR-train | Uniform | **66.8** |
> | | w/o inference | 41.9 |
> | | w/o all | 43.4 |

---

> ### Author Response · Authors · 2025-11-26
> **Rebuttal Followup**
>
> Dear reviewer a385,
>
> As the discussion period ends in less than a week, we wanted to check in to see whether you have any remaining questions. **We would be happy to clarify further, and grateful for any other feedback you may povide**. We really appreciate your time engaged in the review and rebuttal phase.
>
> Thank you very much and look forward to your replies!
>
> Best regards,
>
> Authors of Paper 4205

---

### Official Review · Reviewer_5dth · 2025-10-30

**Soundness:** 2
**Presentation:** 2
**Contribution:** 3
**Rating:** 4
**Confidence:** 4

**Summary:**

This paper introduces VIRTUE, a multimodal embedding model that extends image-text embedding models with visual prompts/instructions. Unlike existing embedding models that only accept text instructions, VIRTUE integrates a segmentation model to process visual prompts (bounding boxes, points, masks) alongside text and images. The authors also contribute SCaR, a new synthetic dataset for evaluating visual-interactive image-to-text retrieval.

**Strengths:**

- They address an existing gap that current models do not support visual prompts. I would, however, love to see this point argued more. E.g., by including examples of real-world uses.
- Contribute a new dataset (SCAR) with challenging negatives
- Their method proposed an elegant adaptation of VLMs for producing embeddings, with image instructions using bounding boxes, which I believe has real-world applications

**Weaknesses:**

- The proposed method, while increasing performance and addressing an existing limitation, does so by increasing model complexity, which might make the model harder to run, optimize, and adapt
- The dataset (SCAR) only explores visual prompts, while I do think that most real-world use cases would use both visual and text prompts. E.g., in the "find similar product" feature, you could mark an item, which then retrieves using the bounding box and a prompt "find products similar to the one highlighted".
- The heavy reliance on GPT-4V and lack of metrics, such as inter-annotator agreement, makes it hard to evaluate the overall quality of the dataset. It is also uncertain how much of the performance gain is learning specific quirks of GPT-4V and how much is generalizable
- I would update figure 2 to include all elements in the filtering (e.g. wordnet)
- Appendix E: Don't name your section "extensive experiments", just call it experiments
- I would argue that the statement: “they only accept text as the human-machine interaction modality” (Anonymous, 2025, p. 1) is borderline incorrect. However, I agree that the model was at least not designed with that intention in mind.
- see question on choice of fine-tuned models

**Questions:**

- why did you choose MMEB over alternative comparable benchmarks
- I am unsure if this model actually supports interactive selection that does not align with the SAM2. Did you experiment with arbitrary interactive human selection?
- Why are only some of the models fine-tuned on MMEB? I am unsure what the selection process was here? Especially interesting it why seemingly competitive models (GME-7B), weren’t fine-tuned.

---

> ### Author Response · Authors · 2025-11-20
> **Response to Reviewer 5dth [1/2]**
>
> Thank you very much for your thoughtful comments, and for suggesting points we can improve on. Your comments are first stated and then followed by our point-to-point responses.
>
> > **[W1] The proposed method increases model complexity, which might make the model harder to run, optimize, and adapt**
>
> The VIRTUE framework does introduce architectural modifications, but they are minor. As acknowledged by `Reviewer a385`, the training process is similar to existing embedding models, and `Reviewer jcPk` recognized the method as a logical combination of SAM and VLM principles.
>
> - Efficiency: Our detailed latency study below (**Appendix E.6**) confirms that the added visual prompt tokens introduce only a negligible overhead in inference time. Additionally, VIRTUE is more efficient than the existing VLM-based embedding models (MMRet, UniME).
> | Model | Query Mode | Time (ms) | Memory (GB) |
> | --- | --- | --- | --- |
> | VIRTUE-7B | w/ visual prompts | 502 | 18.8 |
> | | removing segmentation streamline | 457 | 17.7 |
> | VLM2Vec-7B | N/A | 506 | 18.6 |
> | MMRet-7B | N/A | 638 | 29.0 |
> | UniME-7B | N/A | 519 | 28.8 |
>
> - Optimization  and Adaptability: The optimization process (e.g., hyperparameters, optimizer) of VIRTUE is similar to finetuning the VLM to the embedding model (i.e., without the segmentation streamline), where we did not observe additional instability in practice. Moreover, VIRTUE is architecture-agnostic: replacing Qwen2-VL with Phi-3.5-V required no architectural changes beyond supplying visual prompts in the input sequence. The evaluation results (**Table 9-(1)**) confirm that the method transfers reliably across backbones. We copy the table below as the reference.
> | Backbone | MMEB | SCaR |
> | --- | --- | --- |
> | Phi-3.5-V | 61.7 | 26.5 |
> | Qwen2-VL-2B | 64.8 | 30.4 |
>
> > **[W2] SCaR only explores visual prompts, while most real-world use cases would use both visual and text prompts**
>
> We completely agree and apologize for the confusion in the original manuscript. The SCaR dataset already incorporates text instruction with the visual prompt to simulate real-world compositional queries: "*Find the caption that best describes the segmented object, considering both local details and global context in the given image.*", which is exactly the suggestion from the reviewer. We have updated the manuscript in **Footnote 2 (Page 4)** to clarify this point. We appreciate again for highlighting this insightful comment.
>
> > **[W3.1] Inter-annotator agreement for SCaR.**
>
> We appreciate the reviewer raising this point. To complement the SCaR statistics in Appendix D.4 and the provided samples in the supplementary materials, we have further included an inter-annotator agreement (IAA) study on the evaluation set. With two independent human annotators inspecting the evaluation set, we report the average Cohen’s Kappa below. The high agreement score indicates that the two independent annotators were highly consistent in the dataset inspection process, despite the initial reliance on GPT-4V. We have included this detailed analysis in **Appendix D.4 with Table 7** to substantiate the overall quality of the SCaR dataset.
> | Cohen's Kappa | Drop samples (both annotators flagged) | Drop sampels (either annotator flagger) |
> | --- | --- | --- |
> | 0.89 | 4021 | 1638 |
>
> > **[W3.2] Uncertain how much of the performance gain is learning specific quirks of GPT-4V and how much is generalizable.**
>
> - VIRTUE Generalization: The comparison group of models without SCaR-train (i.e., $\Delta(2B)$ and $\Delta(7B)$ in Table 3) confirms that the initial performance gain is attributable to the visual prompt streamline (our VIRTUE) itself, independent of any potential GPT-4V data bias.
> - SCaR Generalization: As demonstrated in Table 10, adding SCaR-train maintains comparable performance on the universal MMEB benchmarks compared to models not trained on SCaR-train. This stability indicates that our models are not merely learning GPT-4V quirks.
> - Robustness to visual prompts: We have further included Section 5.4 to stress-test the impacts of visual prompts on VIRTUE, which consistently illustrate that our VIRTUE models are robust to noise, misaligned, or various visual prompts (Figure 4, Tables 4 and 5).
>
> This strong generalization and robustness across different benchmarks confirms that VIRTUE is learning robust, generalizable visual-interactive embeddings, not model-specific or stylistic patterns from the data generation process.
>
> > **[W4] Addition of WordNet to Figure 2.**
>
> Thank you for the suggestion. We have added WordNet into the figure accordingly.
>
> > **[W5] Replacement the name of Extensive Experiments in Appendix E.**
>
> Thank you for the feedback! We have revised the section title of Appendix E to "Additional Experiments", which is more accurate and natural, as these results supplement the main experiments presented in the body of the paper.
>
> ---
>
> Please refer to the next block for our responses to W6 and Questions 1-3. Thank you.

---

> ### Author Response · Authors · 2025-11-20
> **Response to Reviewer 5dth [2/2]**
>
> > **[W6] The argument in L44 is borderline incorrect. However, I agree that the model was at least not designed with that intention in mind.**
>
> Thank you for raising this point. We have revised the sentence to state that these models "*rely on text as the primary human-machine interaction modality*" to make the statement more neutral regarding their intended functionality.
>
> > **[Q1] why did you choose MMEB over alternative comparable benchmarks**
>
> We chose MMEB for two primary reasons:
> - Fair Comparison: We follow the established VLM2Vec setting [1], which introduced MMEB as the prevailing benchmark for modern multimodal embedding models and have been widely used by state-of-the-art embedding models (e.g., UniME, MMRet).
> - Universal Scope: While MBEIR [2] focuses narrowly on 8 retrieval tasks across 16 datasets, MMEB serves as a more comprehensive benchmark for evaluating universal embedding ability. MMEB includes 36 datasets spanning four meta-task categories: Classification, VQA, Retrieval, and Visual Grounding.
> The majority of datasets in MBEIR are also included in MMEB. Therefore, MMEB's broader, multi-task coverage allows for a more comprehensive examination of VIRTUE’s universal capabilities across diverse scenarios.
>
> [1] UniIR: Training and Benchmarking Universal Multimodal Information Retrievers. ECCV 2024.
>
> [2] VLM2Vec: Training Vision-Language Models for Massive Multimodal Embedding Tasks. ICLR 2025.
>
> > **[Q2] Did you experiment with arbitrary interactive human selection**
>
> Good question. Yes, since SAM2 was trained with clicks/bboxes/masks (Figure 3 in the SAM2 official paper), it naturally supports arbitrary interactive human selection. We have included additional experiments to test VIRTUE's robustness to other interactive formats:1) Point Prompts: Use only the centroid points of the bounding boxes. 2) Random boxes: Use randomly sampled bounding boxes.
>
> The results confirm that VIRTUE, equipped with the SAM2 backbone, is robust and understandable to variations in visual prompts. Specifically, points and noisy visual prompts led to only a minor drop in performance, indicating that the model is not overfit to perfect bounding box inputs and can generalize well to the imperfect, arbitrary selections typical of human interaction.
> In addition, randomly sampled visual prompts lead to significantly worse results, implying that VIRTUE takes visual prompts into consideration for generating embeddings. We have added this in **Table 4**, and more results on the impacts of noise or misalignment in visual prompts, including jittered boxes, partial masks, and off-by-k pixels, are discussed in **Section 5.4**.
>
> (VIRTUE-7B)
> | | bbox | random | points |
> | --- | --- | --- | --- |
> | SCaR | **27.8** | 9.2 | 26.6 |
>
> (VIRTUE-7B-SCaR-train)
> | | bbox | random | points |
> | --- | --- | --- | --- |
> | SCaR | **56.9** | 13.0 | 45.0 |
>
> > **[Q3] Why are only some of the models fine-tuned on MMEB? I am unsure what the selection process was here?Especially interesting it why seemingly competitive models (GME-7B), weren’t fine-tuned.**
>
> Our selection process for baselines followed a simple and consistent rule: we use **off-the-shelf state-of-the-art results**, including fine-tuned MMEB checkpoints when provided by the original authors. We do not finetune baselines ourselves, as this would introduce discrepancies in training data, preprocessing, and hyperparameters, making the comparison less fair and potentially irreproducible.
>
> For models that were **not** fine-tuned on MMEB (e.g., GME and LamRA), this is because their authors did not release MMEB-fine-tuned versions, and these models were not designed with MMEB’s full task suite in mind. GME and LamRA focus primarily on retrieval or data-synthesis/reasoning, and were trained on retrieval-centric datasets (e.g., LamRA on the MBEIR training set). This aligns with their strong performance on the Retrieval meta-task of MMEB (Table 2), but not necessarily on the other components of MMEB, such as VQA, Classification, or Visual Grounding.
> Therefore, our baseline selection reflects (1) what was publicly available and (2) our goal of comparing VIRTUE fairly against both MMEB-fine-tuned and non-MMEB baselines, without introducing uncontrolled fine-tuning on our side.

---

> ### Author Response · Authors · 2025-11-26
> **Rebuttal Followup**
>
> Dear reviewer 5dth,
>
> As the discussion period ends in less than a week, we wanted to check in to see whether you have any remaining questions. **We would be happy to clarify further, and grateful for any other feedback you may povide**. We really appreciate your time engaged in the review and rebuttal phase.
>
> Thank you very much and look forward to your replies!
>
> Best regards,
>
> Authors of Paper 4205

---

### Official Review · Reviewer_Zhv1 · 2025-11-01

**Soundness:** 3
**Presentation:** 3
**Contribution:** 2
**Rating:** 4
**Confidence:** 3

**Summary:**

The paper proposes VIRTUE, an embedding framework that enhances a frozen vision-language model with a frozen segmentation model (SAM‑2) and a lightweight connector to inject region-level tokens for fine-grained visual interaction. To evaluate this, the authors introduce SCaR, a large benchmark for region-in-context caption retrieval with challenging negatives generated by GPT‑4V. VIRTUE achieves notable gains over baselines on both MMEB and SCaR, with ablations covering alternative encodings and model variants.

**Strengths:**

1. The studied problem is sensible, articulating the gap between text‑only interactions in current embedders and region‑aware visual interactions.
2. A new segmentation-and-scene caption retrieval benchmark is proposed. It also includes case studies for image-to-image retrieval and on-the-fly correction via visual hints, showing great practical value.
3. The experimental results show the proposed framework achieves better results than baseline methods.

**Weaknesses:**

1) The core contribution is essentially feature concatenation from a frozen segmenter into a frozen VLM with a small connector and contrastive fine‑tuning. While effective, there is little architectural insight into how to fuse region tokens with global tokens beyond prepending and projecting; there is no principled modeling of cross‑token interactions (e.g., region‑aware attention, cross‑modal slotting, or routing). This makes the method feel incremental despite the strong results.
2) Baselines that “do not accept boxes” are evaluated primarily via textualizing the bbox or naive cropping; yet there exist stronger region encoders and grounding‑aware alternatives (e.g., leveraging GroundingDINO features, ROIAlign/feature pooling over detected regions, or using grounding‑style LLMs like Ferret) that might close the gap. It is not surprising to me that the proposed method can achieve better performance.
3) After SCaR‑finetuning, VIRTUE‑7B still drops on MMEB (68.6→66.8), while 2B is stable; baselines often degrade more. A deeper analysis is missing.
4) The study varies SSS, SAM‑2 size, and a few hyperparameters, but does not (i) compare frozen vs lightly finetuned SAM‑2, (ii) probe attention patterns to confirm that the LLM actually leverages segmentation tokens for compositional cues, or (iii) test robustness to noisy or misaligned visual prompts (jittered boxes, partial masks, off‑by‑k clicks).
5) Please report end‑to‑end latency and memory for typical query modes (I2T with and without prompts; I2I) versus cropping and text‑only baselines, at matched precision.

**Questions:**

Refer to weakness part.

---

> ### Author Response · Authors · 2025-11-20
> **Response to Reviewer Zhv1 [1/3]**
>
> We thank the reviewer for the constructive comments and positive feedback on our paper. Our point-to-point responses to your comments are summarized below.
>
> > **[W1] No principled modeling of cross token interactions, making the method feel incremental despite the strong results**
>
> We would like to clarify that combining SAM and a VLM into a unified embedding model is non-trivial and introduces several architectural and algorithmic challenges, such as how to design the segmentation-language connector, how to select an appropriate segmentation encoder, how different sizes of SAM affect performance, and how many segmentation tokens should be used. These design questions and their analyses are discussed in **Appendix E.1**. In addition, our core contributions extend beyond the method itself, including the new visual-interactive embedding problem and the empirical validation of its solution, as acknowledged by all reviewers.
>
> To further emphasize our technical contributions, we have conducted new studies to provide clear guidelines and practices for integrating new visual prompt modalities into embedding models, including (1) cross-token interactions (**the table below, Table 9-(8)**), (2) freezing and unfreezing SAM-2 (**Table 9-(9)**), (3) whether VIRTUE really uses segmentation tokens (**Section 5.4 with Table 4**), (4) robustness to visual prompts (**Section 5.4 with Figure 4**), and (5) latency and memory use (**Table 11**). (2-4) can be referred to the response to W4, and (5) is included in the response to W5.
>
> To compare cross-token modeling methods, we implemented and tested two variants for the segmentation-language connector:
> 1) cross-attention: Using the same architecture as the Vision-Language connector in Qwen2-VL.
> 2) attention-pooling: A natural approach using $k$ learnable queries to pool information from the region features (we used $k=64$).
>
> The below results suggest that using attention-based cross-modal methods do not yield better results, although cross-attention achieves comparable performance with using our convolutional-MLPs. We have included this in **Appendix E.1**.
> | Alternative | MMEB | SCaR |
> | --- | --- | --- |
> | Conv-MLP (Ours) | **64.8** | **30.4** |
> | Cross-attention | 63.1 | 30.0 |
> | Attention-pooling | 55.9 | 28.1 |
>
> > **[W2] There exist stronger region encoders and grounding aware alternatives that might close the gap**
>
> We appreciate the reviewer’s suggestion to consider stronger region-aware alternatives. However, our goal is to compare VIRTUE against **embedding models**, and all grounding-aware or detection-oriented models the reviewer suggested do **not** naturally support embedding tasks or visual-interactive retrieval--They are designed for detection or generation tasks. Therefore, their outputs are not aligned to an embedding space, and they cannot produce text–image similarity scores without additional task-specific finetuning. Using them "as-is" would be an unfair comparison because they are not trained for retrieval.
>
> For fairness and reproducibility, we compare VIRTUE against state-of-the-art embedding models (e.g., VLM2Vec, UniME, GME) strictly in their off-the-shelf configurations. To enable region awareness in these models, we only adjust inference-time inputs (e.g., textualizing bounding boxes), avoiding changes to architecture, training data, or hyperparameters that would make comparisons inconsistent or irreproducible.
>
> This gap between region-aware models and embedding models is precisely the motivation for VIRTUE: existing grounding-aware models offer strong regional understanding but are not designed for embedding tasks, while existing embedding models lack region-interactive capabilities entirely.
>
> > **[W3] Missing deeper analysis for the degradance after SCaR-finetuning**
>
> Excellent observation! We also observed these behaviors and have already discussed them in Sec. E.3:
> - All baselines exhibit a degraded performance, likely because the models overfit or shift focus toward the SCaR distribution for incremental learning.
> - VLM2Vec and MMRet suffer severe degradation, indicting stronger susceptibility to catastrophic forgetting since we perform post-finetuning on all models including VIRTUE. In contrast, UniME with multi-stage contrastive learning demonstrates greater robustness.
> - Although VIRTUE-7B shows a minor drop (68.6 to 66.8), its performance remains on par with its original MMEB-trained checkpoints. This stability can be attributed to our segmentation streamline, which enables the use of optional visual prompts and makes VIRTUE more adaptable as a universal embedder.
>
> We are happy to conduct further analysis if the reviewer could provide an explicit direction or specific type of deeper analysis the reviewer would like to see.
>
> ---
> Please refer to the second block for our responses to W4 and the third block for our responses to W5. Thank you.

---

> ### Author Response · Authors · 2025-11-20
> **Response to Reviewer Zhv1 [2/3]**
>
> > **[W4] Additional experiments for (i) freezing and unfeezing SAM-2, (ii) whether the LLM actually leverages segmentation tokens, (iii) robustness to noisy or misaligned visual prompts (jittered boxes, partial masks, off by k clicks).**
>
> Thank you for your valuable suggestions! We have added further experiments to deepen the understanding of how our model incorporates the segmentation streamline. These additional analyses are now included in **Section 5.4**.
>
> (i) Currently, there are no existing LoRA-like methods for lightly finetuning SAM-2. Therefore, we compare full finetuning versus freezing SAM-2 on VIRTUE-2B. As shown in the below table, finetuning SAM-2 performs worse than freezing it, especially on SCaR. This is because MMEB-train does not contain user-specific visual prompts (i.e., it uses uniform points), leading to a mismatch that biases the trainable segmentation model toward non–visual-interactive scenarios. We have included it in **Table 9-(9)**.
> | Finetune SAM-2? | MMEB | SCaR |
> | --- | --- | --- |
> | x (Ours) | 64.8 | 30.4 |
> | v | 63.5 | 25.8 |
>
> (ii) Directly interpreting attention patterns between text tokens and segmentation tokens is challenging because we use segmentation feature maps rather than reconstructed segmentation entities (L311-313). This makes it difficult to assign attention scores to specific image entities automatically. Developing a dedicated method for such interpretation is beyond the scope of this paper.
>
> Instead, we study this behavior empirically: *if the model effectively leverages segmentation tokens, randomly sampling visual prompts should degrade performance; if it does not, performance should remain unchanged or improve* [1, 2]. As shown in the table below, performance significantly drops for SCaR, which requires reasoning on visual prompts--when visual prompts are random. This indicates that our model does leverage segmentation tokens to generate effective embeddings. We have added this in **Table 4**.
>
> (VIRTUE-7B)
> | | bbox | random |
> | --- | --- | --- |
> | SCaR | 27.8 | 9.2 |
>
> (VIRTUE-7B-SCaR-train)
> | | bbox | random |
> | --- | --- | --- |
> | SCaR | 56.9 | 13.0 |
>
> [1] You Mostly Walk Alone: Analyzing Feature Attribution in Trajectory Prediction. ICLR 2022.
>
> [2] ShuttleSHAP: A Turn-Based Feature Attribution Approach for Analyzing Forecasting Models in Badminton. PAKDD 2024.
>
> (iii) We further analyze the impact of jittered boxes, partial masks, and off-by-k pixels on SCaR. Both VIRTUE-7B and VIRTUE-7B-SCaR-train remain stable across varying levels of jitter and offset, demonstrating the robustness of the segmentation streamline. Even when only 20\% of the visual prompt is provided (partial masks), VIRTUE-7B-SCaR-train still outperforms baselines such as VLM2Vec-7B-SCaR-train, showing that our model remains robust even when visual prompts provide limited coverage. We have included those plots in **Figure 4**.
>
> | Jitter fraction | 0\% | 1\% | 10\% | 20\% |
> | --- | --- | --- | --- | --- |
> | VIRTUE-7B | 27.8 | 27.7 | 27.8 | 27.5 |
> | VIRTUE-7B-SCaR-train | 56.9 | 56.8 | 56.5 | 55.0 |
>
> | Partial masking ratio | 20\% | 50\% | 80\% | 100\% |
> | --- | --- | --- | --- | --- |
> | VIRTUE-7B | 25.5 | 25.4 | 27.5 | 27.8 |
> | VIRTUE-7B-SCaR-train | 42.1 | 48.3 | 53.9 | 56.9 |
>
> | Off-by-k pixels | 0 | 10 | 30 | 50 |
> | --- | --- | --- | --- | --- |
> | VIRTUE-7B | 27.8 | 27.6 | 27.4 | 27.3 |
> | VIRTUE-7B-SCaR-train | 56.9 | 54.9 | 54.6 | 53.8 |
>
> Due to the large sizes of including tables, please refer to the third block for W5. Thank you.

---

> ### Author Response · Authors · 2025-11-20
> **Response to Reviewer Zhv1 [3/3]**
>
> > **[W5] Latency and memory for different modes of VIRTUE and baselines**
>
> We appreciate the reviewer's request for a detailed efficiency analysis. We report the average end-to-end latency (ms) and GPU memory (GB) for four distinct operational modes of VIRTUE-7B on the SCaR benchmark (I2T) and provide baselines (VLM2Vec, MMRet, and UniME) for I2I (retrieval and visual grounding) performance from MMEB. For cropping and text-only modes, we remove the visual prompt streamline. All I2T evaluations are reported at matched precision (bf16), and all results are obtained with a batch size of 1 on a single H100 GPU. While cropping and text-only modes offer better latency and memory, the performance becomes significant inferior. We further include the removal of visual prompts for I2I scenarios, which is more efficient but degrading significantly. Moreover, using uniform points is more efficient than the existing baselines. We have included them in **Appendix E.6 and Table 11**.
>
> | | Query Mode | Time (per sample, ms) | Memory (GB) | Performance |
> | --- | --- | --- | --- | --- |
> | I2T (SCaR) | | | | |
> | VIRTUE-7B | w/ visual prompts (Ours) | 422 | 18.8 | 27.8 |
> | | w/o visual prompts (uniform points) | 427 | 18.8 | 16.1 |
> | | Cropping | 139 | 17.7 | 22.8 |
> | | Text-only | 340 | 17.7 | 23.9 |
> | I2I (Retrieval and Visual Grounding on MMEB) | | | | |
> | VIRTUE-7B | w/ uniform points (Ours) | 402 | 18.8 | 70.7 |
> | | w/o uniform points (remove the segmentation streamline) | 357 | 17.7 | 46.5 |
> | VLM2Vec-7B | N/A | 406 | 18.6 | 67.9 |
> | MMRet-7B | N/A | 538 | 29.0 | 75.8 |
> | UniME | N/A | 419 | 28.9 | 66.4

---

> ### Author Response · Authors · 2025-11-26
> **Rebuttal followup**
>
> Dear reviewer Zhv1,
>
> As the discussion period ends in less than a week, we wanted to check in to see whether you have any remaining questions. **We would be happy to clarify further, and grateful for any other feedback you may povide**. We really appreciate your time engaged in the review and rebuttal phase.
>
> Thank you very much and look forward to your replies!
>
> Best regards,
>
> Authors of Paper 4205

---

### Author Response · Authors · 2025-11-20
**Summary of Revision**

We sincerely appreciate all reviewers for their time and for their constructive feedback with our work. We are encouraged that **all reviewers** recognized that our work addresses an important gap: current embedding models do not support visual prompts that **compliment to textual prompts and are practical in real-world applications** (`Reviewer a385`), thereby **expanding the possibilities for human-computer interaction** (`Reviewer jcPk`).

We are also excited that **all reviewers** acknowledged our contributions from the **challenging** (`Reviewer 5dth`), **beneficial** (`Reviewer a385`), **valuable** (`Reviewer jcPk`), proposed benchmark as well as **effective** (`Reviewers Zhv1`, `a385`) and **comprehensive** (`Reviewer jcPk`) experimental results. In particular, `Reviewer Zhv1` highlighted our *"case studies on image-to-image retrieval and on-the-fly correction via visual hints, showing great practical value''* .

Finally, we appreciate that the reviewers characterized our proposed method as **beneficial** (`Reviewer a385`), **an elegant adaptation of VLMs for producing embeddings with image instructions** (`Reviewer 5dth`), and a method that **overcomes the limitations of traditional methods that exclusively support textual prompts** (`Reviewer jcPk`).

---

To address the reviewers’ questions and concerns, we have revised the manuscript accordingly. All updates are highlighted in **bold blue**, and the same additions are reflected in our detailed responses.
- Additional experiments for guidelines and practices (`Reviewer Zhv1`): We have included the results for (1) different segmentation-language connectors in **Table 9-(8)** and (2) freezing and unfreezing SAM-2 in **Table 9-(9)** to provide clearer guidance on incorporating visual prompts into VLM embedding models. The corresponding discussions are included in **Appendix E.1**.
- Impacts of various visual prompts on VIRTUE (`Reviewers Zhv1, a385, 5dth`): We have added **Section 5.4** to analyze robustness to noisy or misaligned visual prompts (**Figure 4**), the effects of points, bounding boxes, and random boxes to verify our model really leverages segmentation tokens (**Table 4**), and the impacts of removing uniformly sampled points on MMEB (**Figure 5**). We have also updated **L110-111** to further emphasize our method’s contribution accordingly.
- Latency and memory analysis (`Reviewer Zhv1`): We have reported end-to-end inference time and memory for different query modes of VIRTUE and the baselines, under both non-visual-interactive and visual-interactive scenarios in **Appendix E.6** and **Table 11**.
- Inter-annotator agreement on SCaR (`Reviewer 5dth`): We have added **Table 7** and descriptions in **Appendix D.4**, including agreement statistics and the numbers of drop samples in building SCaR.
- Manuscript clarifications (`Reviewer 5dth`):
	- We have added **Footnote 2 (Page 4)** to clarify that SCaR inputs contain textual instructions.
	- We have updated **Figure 2** to include WordNet in the filtering pipeline.
	- We have revised the **section title of Appendix E** and the statement in **L44-45** for a more neutral tone.

Once again, we would like to thank all the reviewers for their constructive suggestions, which have helped us improve our work.

---

### Author Response · Authors · 2025-11-29
**Summary of Responses for Paper 4205**

Dear (new) ACs and SACs,

We greatly appreciate your efforts in coordinating our submission under these unusual circumstances. While we submitted our responses on **Nov. 20**, the discussion period closed before receiving any reviewer feedback. In light of this, we provide a one-sentence summary for each response. All experiments and clarifications are added, addressing reviewer concerns on novelty, prompts, and datasets. Please refer to *Summary of Revision* and the full responses for details.

> **[Reviewers a385 (W2), jcPk (W1)] Limited technical novelty**

We clarified the non-trivial architectural challenges of integrating SAM and VLMs and added new ablations on cross-token interactions (**Tab. 9-(8), App. E.1**), SAM-2 freezing vs. unfreezing (**Tab. 9-(9)**), segmentation-token usage (**Tab. 4, Sec. 5.4**), robustness to visual prompts (**Fig. 4, Sec. 5.4**), and latency and memory use (**Tab. 11, App. E.6**).

## Reviewer Zhv1
> **[W1] Whether the method lacks a principled cross-token modeling design.**

We detailed the architectural challenges in integrating SAM and VLMs and added new ablations on cross-token interactions (**Tab. 9-(8), App. E.1**), SAM-2 freezing vs. unfreezing (**Tab. 9-(9)**), segmentation-token usage (**Tab. 4, Sec. 5.4**), robustness to visual prompts (**Fig. 4, Sec. 5.4**), and latency and memory use (**Tab. 11, App. E.6**).

> **[W2] Suggested that grounding-aware models might close the gap.**

We explained that grounding models are not embedding models and cannot be used for retrieval without task-specific finetuning; thus, we follow fair, reproducible, embedding-model baselines consistent with our problem scope.

> **[W3] Deeper analysis of the performance drop after SCaR-finetuning.**

We also observed them and have *already* discussed them in **App. E.3**.

> **[W4, W5] Experiments on SAM-2 finetuning, segmentation-token usage, robustness to noisy prompts, and efficient use.**

As summarized in W1, we added experiments showing freezing SAM-2 is superior, segmentation tokens are used (random prompts cause strong drops), the model is robust to jitter, partial masks, and positional offsets, and our model is efficient with the strongest accuracy.

## Reviewer 5dth
> **[W1] Worried about increased complexity**

Our added components introduce only negligible runtime overhead, maintain stable optimization, and transfer cleanly across backbones, as verified by latency studies (**App. E.6**) and cross-model evaluations (**Tab. 9-(1)**).

> **[W2] SCaR uses only visual prompts**

SCaR already incorporates text instructions alongside visual prompts, and we clarified this explicitly in the revised manuscript (**Footnote 2**).

> **[W3.1] Inter-annotator agreement for SCaR**

We added an IAA study showing high consistency (Cohen’s Kappa=0.89) between two independent annotators, now included in **App. D.4**.

> **[W3.2] Learn from GPT-4V quirks vs. generalization**

Performance gains stem from VIRTUE’s design rather than GPT-4V biases, confirmed by VIRTUE-only baselines (**Tab. 3**), stable MMEB performance with SCaR-train (**Tab. 10**), and new robustness tests (**Fig. 4, Tab. 4 & 5**).

> **[W4, W5, W6] Paper Improvements**

We have revised them accordingly.

> **[Q1] Why MMEB over other benchmarks**

We chose MMEB since it is the standard universal embedding benchmark with broader and more diverse coverage than MBEIR.

> **[Q2] Arbitrary interactive human selection**

We tested points, noisy boxes, and random boxes (**Sec. 5.4**), finding VIRTUE robust to realistic interactions and sensitive to randomized prompts, confirming meaningful prompt use.

> **[Q3] Why some baselines are not MMEB-finetuned**

We use only author-released checkpoints for fairness and reproducibility, and models like GME/LamRA do *not* provide MMEB-finetuned versions.

## Reviewer a385
> **[W1] Narrow scope/caption-retrieval-only benchmark**

Although SCaR is formatted as caption retrieval, its structured hard negatives and multi-dataset design make it a compositional and visual-reasoning benchmark that *complements* general-purpose suites like MMEB.

> **[W3] Validate on more MLLM families/including GME-Qwen**

Our results *already* include GME-Qwen and multiple MLLM families (Qwen2-VL, LLaVA-1.6, LLaVA-NeXT), as shown in **Tab. 2 and 3**.

> **[W4] Drop visual prompts for efficiency**

We added experiments by removing visual prompts either only at inference or in both stages (**Tab. 5**), showing that it significantly degrades accuracy, confirming their necessity despite small efficiency gains.

## Reviewer jcPk
> **[Q1] Contribution of dataset vs. method**

**Tab. 3 and 7** *already* show relative contributions of both our VIRTUE and SCaR.

> **[Q2] Bias or semantic repetition in SCaR negatives**

We *already* addressed synonym/hypernym repetition by prompting and WordNet-based filtering (**Fig. 4**) for diverse and unbiased hard negatives.

We sincerely thank the ACs and hope this summary aids evaluation.

---

> ### Author Response · Authors · 2025-11-29
>
> We believe the revised manuscript and responses **address all raised concerns**, and we are happy to clarify any remaining questions during the final days of the rebuttal process.
>
> Authors of Paper 4205

---

### Meta-Review · Area_Chair_NUCx · 2026-01-07

**Summary:**

Reviewers generally agreed that the paper addresses a meaningful and previously underexplored problem—enabling visual interaction in multimodal embedding models—and that the empirical results on both MMEB and the proposed SCaR benchmark are strong. The main concerns focused on (i) limited architectural novelty, with the method perceived as a relatively incremental integration of a segmentation model and a VLM, (ii) the scope and synthetic nature of the SCaR benchmark, and (iii) practical questions around efficiency, robustness, and dataset quality.

The rebuttal substantially addressed most of these issues by adding extensive ablations and analyses, including efficiency and memory studies, robustness to noisy or misaligned visual prompts, evidence that segmentation tokens are meaningfully used, comparisons of freezing vs. finetuning SAM-2, and improved validation of the SCaR dataset (e.g., inter-annotator agreement and bias filtering). These additions resolve the major practical and empirical concerns raised by the reviewers.

Some reviewers still view the core fusion design as architecturally incremental, and the benchmark as complementary rather than fully universal. However, these remaining concerns relate primarily to the degree of conceptual novelty rather than to soundness or empirical support. Balancing the clarified contributions, strong experimental validation, and practical relevance, I recommend accept (poster).

**Reviewer Concerns:**

Reviewer Concerns

The rebuttal addressed most practical and empirical concerns raised by the reviewers, but some conceptual concerns remain.

Concerns largely addressed:
Issues around efficiency and added complexity, robustness to noisy or misaligned visual prompts, whether segmentation tokens are actually used, freezing vs. finetuning SAM-2, dataset quality and reliability (inter-annotator agreement, GPT-4V bias, WordNet filtering), benchmark choice (MMEB), and the separation between method and dataset contributions were convincingly addressed with new experiments, analyses, and clarifications. These responses are sufficient to resolve the main concerns of Reviewer 5dth and to clarify the questions raised by Reviewers jcPk and a385.

Concerns partially outstanding:
A recurring concern from Reviewers Zhv1 and a385 is that the core model design remains incremental, relying on a lightweight connector rather than a more principled cross-token or region-aware fusion mechanism. While the authors added comparative ablations and diagnostics, this concern is only partially mitigated and remains a matter of perceived architectural novelty rather than empirical validity.

Overall, the rebuttal substantially strengthened the paper by resolving most major technical and experimental concerns; the remaining issues mainly relate to the degree of architectural novelty, not to soundness or evaluation.

**Reviewer Scores:**

Reviewer Scores (expected after full discussion)

Reviewer jcPk: 6 → 6
The rebuttal clarifies method vs. dataset contributions and addresses concerns about negative-sample bias, but the reviewer’s main stance (“limited technical innovation”) likely remains. Net effect: clearer, but not enough to justify a score bump.

Reviewer a385: 6 → 6
The rebuttal strengthens empirical support (efficiency, prompt necessity, broader baseline coverage), but the core novelty/scope concerns are only partially softened (still feels like a SAM+VLM combination; SCaR still narrower than universal suites). Likely stays.

Reviewer 5dth: 4 → 6
Major practical concerns (added complexity/efficiency, dataset quality via IAA, real-world prompt setting, and robustness) are addressed with concrete evidence and clarifications. Even without an explicit reviewer reply, the rebuttal is strong enough to justify a +2.

Reviewer Zhv1: 4 → 4
Several requested experiments are added (robustness, efficiency, freezing vs. finetuning, evidence of using segmentation tokens), but the reviewer’s central “major” concern—lack of a more principled cross-token fusion design and stronger region-aware comparisons—still partially stands. Under the guideline of not bumping when a major concern remains, I’d keep the score unchanged.

Summary: Expected post-discussion scores = [6, 6, 6, 4]
Mean: 5.5

---

### Decision · Program_Chairs · 2026-01-26

Accept (Poster)